# ADAMP: SLOWING DOWN THE SLOWDOWN FOR MO- MENTUM OPTIMIZERS ON SCALE-INVARIANT WEIGHTS

**Byeongho Heo**[*,1]  **Sanghyuk Chun**[*,1,2]  **Seong Joon Oh**[1]  **Dongyoon Han**[1]
**Sangdoo Yun**[1]  **Gyuwan Kim**[1,2]  **Youngjung Uh**[3,†]  **Jung-Woo Ha**[1,2]

Naver AI Lab[1], Naver Clova[2]
Applied Information Engineering, Yonsei University[3]
Equal contribution[*], Works done at Naver AI Lab[†]

## ABSTRACT

Normalization techniques, such as batch normalization (BN), are a boon for modern deep learning. They let weights converge more quickly with often better generalization performances. It has been argued that the normalization-induced scale invariance among the weights provides an advantageous ground for gradient descent (GD) optimizers: the effective step sizes are automatically reduced over time, stabilizing the overall training procedure. It is often overlooked, however, that the additional introduction of *momentum* in GD optimizers results in a far more rapid reduction in effective step sizes for scale-invariant weights, a phenomenon that has not yet been studied and may have caused unwanted side effects in the current practice. This is a crucial issue because arguably the vast majority of modern deep neural networks consist of (1) momentum-based GD (*e.g.* SGD or Adam) and (2) scale-invariant parameters (*e.g.* more than 90% of the weights in ResNet are scale-invariant due to BN). In this paper, we verify that the widely-adopted combination of the two ingredients lead to the premature decay of effective step sizes and sub-optimal model performances. We propose a simple and effective remedy, SGDP and AdamP: get rid of the radial component, or the norm-increasing direction, at each optimizer step. Because of the scale invariance, this modification only alters the effective step sizes without changing the effective update directions, thus enjoying the original convergence properties of GD optimizers. Given the ubiquity of momentum GD and scale invariance in machine learning, we have evaluated our methods against the baselines on 13 benchmarks. They range from vision tasks like classification (*e.g.* ImageNet), retrieval (*e.g.* CUB and SOP), and detection (*e.g.* COCO) to language modelling (*e.g.* WikiText) and audio classification (*e.g.* DCASE) tasks. We verify that our solution brings about uniform gains in performances in those benchmarks. Source code is available at https://github.com/clovaai/adamp.

## 1 INTRODUCTION

Normalization techniques, such as batch normalization (BN) (Ioffe & Szegedy, 2015), layer normalization (LN) (Ba et al., 2016), instance normalization (IN) (Ulyanov et al., 2016), and group normalization (GN) (Wu & He, 2018), have become standard tools for training deep neural network models. Originally proposed to reduce the internal covariate shift (Ioffe & Szegedy, 2015), normalization methods have proven to encourage several desirable properties in deep neural networks, such as better generalization (Santurkar et al., 2018) and the scale invariance (Hoffer et al., 2018).

Prior studies have observed that the normalization-induced scale invariance of weights stabilizes the convergence for the neural network training (Hoffer et al., 2018; Arora et al., 2019; Kohler et al., 2019; Dukler et al., 2020). We provide a sketch of the argument here. Given weights $\boldsymbol{w}$ and an input $\boldsymbol{x}$, we observe that the normalization makes the weights become scale-invariant:

$$\text{Norm}(\boldsymbol{w}^\top \boldsymbol{x}) = \text{Norm}(c\boldsymbol{w}^\top \boldsymbol{x}) \quad \forall c > 0. \tag{1}$$

The resulting equivalence relation among the weights lets us consider the weights only in terms of their $\ell_2$-normalized vectors $\widehat{\boldsymbol{w}} := \frac{\boldsymbol{w}}{\|\boldsymbol{w}\|_2}$ on the sphere $\mathbb{S}^{d-1} = \{\boldsymbol{v} \in \mathbb{R}^d \; : \; \|v\|_2 = 1\}$. We refer to

$\mathbb{S}^{d-1}$ as the *effective space*, as opposed to the nominal space $\mathbb{R}^d$ where the actual optimization algorithms operate. The mismatch between these spaces results in the discrepancy between the gradient descent steps on $\mathbb{R}^d$ and their effective steps on $\mathbb{S}^{d-1}$. Specifically, for the gradient descent updates, the *effective step sizes* $\|\Delta\widehat{\boldsymbol{w}}_{t+1}\|_2 := \|\widehat{\boldsymbol{w}}_{t+1} - \widehat{\boldsymbol{w}}_t\|_2$ are the scaled versions of the nominal step sizes $\|\Delta\boldsymbol{w}_{t+1}\|_2 := \|\boldsymbol{w}_{t+1} - \boldsymbol{w}_t\|_2$ by the factor $\frac{1}{\|\boldsymbol{w}_t\|_2}$ (Hoffer et al., 2018). Since $\|\boldsymbol{w}_t\|_2$ increases during training (Soudry et al., 2018; Arora et al., 2019), the effective step sizes $\|\Delta\widehat{\boldsymbol{w}}_t\|_2$ decrease as the optimization progresses. The automatic decrease in step sizes stabilizes the convergence of gradient descent algorithms applied on models with normalization layers: even if the nominal learning rate is set to a constant, the theoretically optimal convergence rate is guaranteed (Arora et al., 2019).

In this work, we show that the widely used *momentum*-based gradient descent optimizers (*e.g.* SGD and Adam (Kingma & Ba, 2015)) decreases the effective step size $\Delta\widehat{\boldsymbol{w}}_t$ even more rapidly than the momentum-less counterparts considered in (Arora et al., 2019). This leads to a slower effective convergence for $\widehat{\boldsymbol{w}}_t$ and potentially sub-optimal model performances. We illustrate this effect on a 2D toy optimization problem in Figure 1. Compared to "GD", "GD+momentum" is much faster in the nominal space $\mathbb{R}^2$, but the norm growth slows down the effective convergence in $\mathbb{S}^1$, reducing the acceleration effect of momentum. This phenomenon is not confined to the toy setup, for example, 95.5% and 91.8% of the parameters of the widely-used ResNet18 and ResNet50 (He et al., 2016) are scale-invariant due to BN. The majority of deep models nowadays are trained with SGD or Adam with momentum. And yet, our paper is first to delve into the issue in the widely-used combination of scale-invariant parameters and momentum-based optimizers.

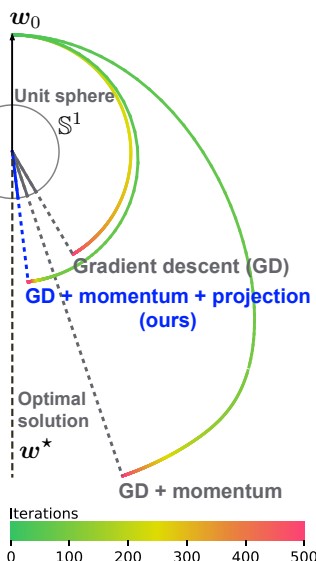

Figure 1. **Optimizer trajectories.** Shown is the $\boldsymbol{w}_t$ for the optimization problem $\max_{\boldsymbol{w}} \frac{\boldsymbol{w}^\top \boldsymbol{w}^\star}{\|\boldsymbol{w}\|_2 \|\boldsymbol{w}^\star\|_2} s$. Trajectories start from $\boldsymbol{w}_0$ towards the optimal solution $\boldsymbol{w}^\star$. The problem is invariant to the scale of $\boldsymbol{w}$. Video version in the attached code.

We propose a simple solution to slow down the decay of effective step sizes while maintaining the step directions of the original optimizer in the effective space. At each iteration of a momentum-based gradient descent optimizer, we propose to project out the radial component (*i.e.* component parallel to $\boldsymbol{w}$) from the update, thereby reducing the increase in the weight norm over time. Because of the scale invariance, the procedure does not alter the update direction in the effective space; it only changes the effective step sizes. We can observe the benefit of our optimizer in the toy setting in Figure 1. "Ours" suppresses the norm growth and thus slows down the effective learning rate decay, allowing the momentum-accelerated convergence in $\mathbb{R}^2$ to be transferred to the actual space $\mathbb{S}^1$. "Ours" converges most quickly and achieves the best terminal objective value. We do not discourage the use of momentum-based optimizers; momentum is often an indispensable ingredient that enables best performances by deep neural networks. Instead, we propose to use our method that helps momentum realize its full potential by letting the acceleration operate on the effective space, rather than squandering it on increasing norms to no avail.

The projection algorithm is simple and readily applicable to various optimizers for deep neural networks. We apply this technique on SGD and Adam (SGDP and AdamP, respectively) and verify the slower decay of effective learning rates as well as the resulting performance boosts over a diverse set of practical machine learning tasks including image classification, image retrieval, object detection, robustness benchmarks, audio classification, and language modelling.

As a side note, we have identified certain similarities between our approaches and Cho & Lee (2017). Cho & Lee (2017) have considered performing the optimization steps for the scale-invariant parameters on the spherical manifold. We argue that our approaches are conceptually different, as ours operate on the ambient Euclidean space, and are more practical. See Appendix §G.1 for a more detailed argumentation based on conceptual and empirical comparisons.

## 2 PROBLEM

Widely-used normalization techniques (Ioffe & Szegedy, 2015; Salimans & Kingma, 2016; Ba et al., 2016; Ulyanov et al., 2016; Wu & He, 2018) in deep networks result in the scale invariance for

weights. We show that the introduction of momentum in gradient-descent (GD) optimizers, when applied on such scale-invariant parameters, decreases the effective learning rate much more rapidly. This phenomenon has not yet been studied in literature, despite its ubiquity. We suspect the resulting early convergence may have introduced sub-optimality in many SGD and Adam-trained models across machine learning tasks. The analysis motivates our optimizer in §3.

## 2.1 Normalization layer and scale invariance

For a tensor $x \in \mathbb{R}^{n_1 \times \cdots \times n_r}$ of rank $r$, we define the normalization operation along the axes $\boldsymbol{k} \in \{0,1\}^{\{1,\cdots,r\}}$ as $\mathrm{Norm}_{\boldsymbol{k}}(x) = \frac{x - \mu_{\boldsymbol{k}}(x)}{\sigma_{\boldsymbol{k}}(x)}$ where $\mu_{\boldsymbol{k}}, \sigma_{\boldsymbol{k}}$ are the mean and standard deviation functions along the axes $\boldsymbol{k}$, without axes reduction (to allow broadcasted operations with $x$). Depending on $\boldsymbol{k}$, $\mathrm{Norm}_{\boldsymbol{k}}$ includes special cases like batch normalization (BN) (Ioffe & Szegedy, 2015).

For a function $g(\boldsymbol{u})$, we say that $g$ is **scale invariant** if $g(c\boldsymbol{u}) = g(\boldsymbol{u})$ for any $c > 0$. We then observe that $\mathrm{Norm}(\cdot)$ is scale invariant. In particular, under the context of neural networks,

$$\mathrm{Norm}(\boldsymbol{w}^\top \boldsymbol{x}) = \mathrm{Norm}((c\boldsymbol{w})^\top \boldsymbol{x}) \tag{2}$$

for any $c > 0$, leading to the scale invariance against the weights $\boldsymbol{w}$ preceding the normalization layer. The norm of such weights $\|\boldsymbol{w}\|_2$ does not affect the forward $f_{\boldsymbol{w}}(\boldsymbol{x})$ or the backward $\nabla_{\boldsymbol{w}} f_{\boldsymbol{w}}(\boldsymbol{x})$ computations of a neural network layer $f_{\boldsymbol{w}}$ parameterized by $\boldsymbol{w}$. We may represent the **scale-invariant weights** via their $\ell_2$-normalized vectors $\widehat{\boldsymbol{w}} := \frac{\boldsymbol{w}}{\|\boldsymbol{w}\|_2} \in \mathbb{S}^{d-1}$ (*i.e.* $c = \frac{1}{\|\boldsymbol{w}\|_2}$).

## 2.2 Notations for the optimization steps

See the illustration on the right for the summary of notations describing an optimization step. We write a gradient descent (GD) algorithm as:

$$\boldsymbol{w}_{t+1} \leftarrow \boldsymbol{w}_t - \eta \boldsymbol{p}_t \tag{3}$$

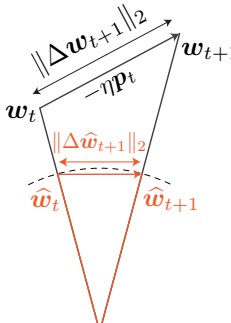

where $\eta > 0$ is the user-defined **learning rate**. The norm of the difference $\|\Delta \boldsymbol{w}_{t+1}\|_2 := \|\boldsymbol{w}_{t+1} - \boldsymbol{w}_t\|_2 = \eta \|\boldsymbol{p}_t\|_2$ is referred to as the **step size**. When $\boldsymbol{p} = \nabla_{\boldsymbol{w}} f(\boldsymbol{w})$, equation 3 is the vanilla GD algorithm. Momentum-based variants have more complex forms for $\boldsymbol{p}$.

In this work, we study the optimization problem in terms of the $\ell_2$ normalized weights in $\mathbb{S}^{d-1}$, as opposed to the nominal space $\mathbb{R}^d$. As the result of equation 3, an **effective optimization step** takes place in $\mathbb{S}^{d-1}$: $\Delta \widehat{\boldsymbol{w}}_{t+1} := \widehat{\boldsymbol{w}}_{t+1} - \widehat{\boldsymbol{w}}_t$. We refer to the **effective step size** $\|\Delta \widehat{\boldsymbol{w}}_{t+1}\|_2$.

## 2.3 Effective step sizes for vanilla gradient descent (GD)

We approximate the effective step sizes for the scale-invariant $\boldsymbol{w}$ under the vanilla GD algorithm. We observe that the scale invariance $f(c\boldsymbol{w}) \equiv f(\boldsymbol{w})$ leads to the orthogonality:

$$0 = \frac{\partial f(c\boldsymbol{w})}{\partial c} = \boldsymbol{w}^\top \nabla_{\boldsymbol{w}} f(\boldsymbol{w}). \tag{4}$$

For example, the vanilla GD update step $\boldsymbol{p} = \nabla_{\boldsymbol{w}} f(\boldsymbol{w})$ is always perpendicular to $\boldsymbol{w}$. Based on this, we establish the effective step size for $\boldsymbol{w}$ on $\mathbb{S}^{d-1}$:

$$\|\Delta \widehat{\boldsymbol{w}}_{t+1}\|_2 := \left\| \frac{\boldsymbol{w}_{t+1}}{\|\boldsymbol{w}_{t+1}\|_2} - \frac{\boldsymbol{w}_t}{\|\boldsymbol{w}_t\|_2} \right\|_2 \approx \left\| \frac{\boldsymbol{w}_{t+1}}{\|\boldsymbol{w}_{t+1}\|_2} - \frac{\boldsymbol{w}_t}{\|\boldsymbol{w}_{t+1}\|_2} \right\|_2 = \frac{\|\Delta \boldsymbol{w}_{t+1}\|_2}{\|\boldsymbol{w}_{t+1}\|_2} \tag{5}$$

where the approximation assumes $\frac{1}{\|\boldsymbol{w}_{t+1}\|_2} - \frac{1}{\|\boldsymbol{w}_t\|_2} = o(\eta)$, which holds when $\boldsymbol{p}_t \perp \boldsymbol{w}_t$ as in the vanilla GD. We have thus derived that the effective step size on $\mathbb{S}^{d-1}$ is inversely proportional to the weight norm, in line with the results in Hoffer et al. (2018).

Having established the relationship between the effective step sizes and the weight norm of a scale-invariant parameters (*s.i.p*), we derive the formula for its growth under the vanilla GD optimization.

**Lemma 2.1** (Norm growth by GD, Lemma 2.4 in Arora et al. (2019)). *For a s.i.p. $\boldsymbol{w}$ and the vanilla GD, where $\boldsymbol{p}_t = \nabla_{\boldsymbol{w}} f(\boldsymbol{w}_t)$,*

$$\|\boldsymbol{w}_{t+1}\|_2^2 = \|\boldsymbol{w}_t\|_2^2 + \eta^2 \|\boldsymbol{p}_t\|_2^2. \tag{6}$$

The lemma follows from the orthogonality in equation 4. It follows that the norm of a scale-invariant parameter $\|\boldsymbol{w}\|_2$ is monotonically increasing and consequently decreases the effective step size for $\boldsymbol{w}$. Arora et al. (2019) has further shown that GD with the above adaptive step sizes converges to a stationary point at the theoretically optimal convergence rate $O(T^{-1/2})$ under a fixed learning rate.

### 2.4 Rapid decay of effective step sizes for momentum-based GD

Momentum is designed to accelerate the convergence of gradient-based optimization by letting $\boldsymbol{w}$ escape high-curvature regions and cope with small and noisy gradients. It has become an indispensable ingredient for training modern deep neural networks. A momentum update follows:

$$\boldsymbol{w}_{t+1} \leftarrow \boldsymbol{w}_t - \eta \boldsymbol{p}_t, \qquad \boldsymbol{p}_t \leftarrow \beta \boldsymbol{p}_{t-1} + \nabla_{\boldsymbol{w}_t} f(\boldsymbol{w}_t) \tag{7}$$

for steps $t \geq 0$, where $\beta \in (0, 1)$ and $\boldsymbol{p}_{-1}$ is initialized at $\mathbf{0}$. Note that the step direction $\boldsymbol{p}_t$ and the parameter $\boldsymbol{w}_t$ may not be perpendicular anymore. We show below that momentum increases the weight norm under the scale invariance, even more so than does the vanilla GD.

**Lemma 2.2** (Norm growth by momentum). *For a s.i.p. $\boldsymbol{w}$ updated via equation 7, we have*

$$\|\boldsymbol{w}_{t+1}\|_2^2 = \|\boldsymbol{w}_t\|_2^2 + \eta^2 \|\boldsymbol{p}_t\|_2^2 + 2\eta^2 \sum_{k=0}^{t-1} \beta^{t-k} \|\boldsymbol{p}_k\|_2^2. \tag{8}$$

Proof is in the Appendix §A. Comparing Lemma 2.1 and 2.2, we notice that the formulation is identical, except for the last term on the right hand side of Lemma 2.2. This term is not only non-negative, but also is an accumulation of the past updates. This additional term results in the significantly accelerated increase of weight norms when the momentum is used. We derive a more precise asymptotic ratio of the weight norms for the GD with and without momentum below.

**Corollary 2.3** (Asymptotic norm growth comparison). *Let $\|\boldsymbol{w}_t^{\text{GD}}\|_2$ and $\|\boldsymbol{w}_t^{\text{GDM}}\|_2$ be the weight norms at step $t \geq 0$, following the recursive formula in Lemma 2.1 and 2.2, respectively. We assume that the norms of the updates $\|\boldsymbol{p}_t\|_2$ for GD with and without momentum are identical for every $t \geq 0$. We further assume that the sum of the update norms is non-zero and bounded: $0 < \sum_{t \geq 0} \|\boldsymbol{p}_t\|_2^2 < \infty$. Then, the asymptotic ratio between the two norms is given by:*

$$\frac{\|\boldsymbol{w}_t^{\text{GDM}}\|_2^2 - \|\boldsymbol{w}_0\|_2^2}{\|\boldsymbol{w}_t^{\text{GD}}\|_2^2 - \|\boldsymbol{w}_0\|_2^2} \longrightarrow 1 + \frac{2\beta}{1-\beta} \quad \text{as} \quad t \to \infty. \tag{9}$$

Proof in the Appendix §A. While the identity assumption for $\|\boldsymbol{p}_t\|_2$ between GD with and without momentum is strong, the theory is designed to illustrate an approximate norm growth ratios between the algorithms. For a popular choice of $\beta = 0.9$, the factor is as high as $1 + 2\beta/(1 - \beta) = 19$. Our observations are also applicable to Nesterov momentum and momentum-based adaptive optimizers like Adam. We later verify that the momentum induce the increase in weight norms and thus rapidly reduces the effective learning rates in many realistic setups of practical relevance (§3.2 and §4).

## 3 Method

We have studied the accelerated decay of effective learning rates for scale-invariant weights (*e.g.* those preceding a normalization layer) under the momentum. In this section, we propose a projection-based solution that prevents the momentum-induced effective step size decreases while not changing the update directions in the effective weight space $\mathbb{S}^{d-1}$.

### 3.1 Our method: Projected updates

We remove the accumulated error term in Lemma 2.2, while retaining the benefits of momentum, through a simple modification. Let $\Pi_{\boldsymbol{w}}(\cdot)$ be a projection onto the tangent space of $\boldsymbol{w}$:

$$\Pi_{\boldsymbol{w}}(\boldsymbol{x}) := \boldsymbol{x} - (\widehat{\boldsymbol{w}} \cdot \boldsymbol{x})\widehat{\boldsymbol{w}}. \tag{10}$$

We apply $\Pi_{\boldsymbol{w}}(\cdot)$ to the momentum update $\boldsymbol{p}$ (equation 7) to remove the radial component, which accumulates the weight norms without

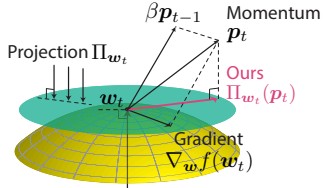

Figure 2. Vector directions of the gradient, momentum, and ours.

contributing to the optimization. Our modified update rule is:

$$\boldsymbol{w}_{t+1} = \boldsymbol{w}_t - \eta \boldsymbol{q}_t,$$

$$\boldsymbol{q}_t = \begin{cases} \Pi_{\boldsymbol{w}_t}(\boldsymbol{p}_t) & \text{if } \cos(\boldsymbol{w}_t, \nabla_{\boldsymbol{w}} f(\boldsymbol{w}_t)) < \delta/\sqrt{\dim(\boldsymbol{w})} \\ \boldsymbol{p}_t & \text{otherwise} \end{cases} \tag{11}$$

where $\cos(\boldsymbol{a}, \boldsymbol{b}) := \frac{|\boldsymbol{a}^\top \boldsymbol{b}|}{\|\boldsymbol{a}\|\|\boldsymbol{b}\|}$ is the cosine similarity. Instead of manually registering weights preceding normalization layers, our algorithm automatically detects scale invariances with the cosine similarity for user convenience. In all experiments considered, we found $\delta = 0.1$ to be sufficiently small to precisely detect orthogonality and sufficiently large to recall all scale-invariant weights (Appendix §C): we suggest future users to use the same value. The proposed update rule makes a scale-invariant parameter $\boldsymbol{w}$ perpendicular to its update step $\boldsymbol{q}$. It follows then that the rapid weight norm accumulation shown in Lemma 2.2 is alleviated back to the vanilla gradient descent growth rate in Lemma 2.1 due to the orthogonality:

$$\|\boldsymbol{w}_{t+1}\|_2^2 = \|\boldsymbol{w}_t\|_2^2 + \eta^2 \|\boldsymbol{q}_t\|_2^2 \leq \|\boldsymbol{w}_t\|_2^2 + \eta^2 \|\boldsymbol{p}_t\|_2^2 \tag{12}$$

where inequality follows from the fact that $\boldsymbol{q}_t = \Pi_{\boldsymbol{w}_t}(\boldsymbol{p}_t)$ and $\Pi_{\boldsymbol{w}_t}$ is a projection operation. Although the updates $\boldsymbol{p}_t$ are not identical between equation 8 and equation 12, we observe that after our orthogonal projection, the updates no longer get accumulated as in the last term of equation equation 8. We emphasize that this modification only alters the effective learning rate while not changing the effective update directions, as shown in the below proposition.

**Proposition 3.1** (Effective update direction after projection). *Let $\boldsymbol{w}_{t+1}^o := \boldsymbol{w}_t - \eta \boldsymbol{p}_t$ and $\boldsymbol{w}_{t+1}^p := \boldsymbol{w}_t - \eta \Pi_{\boldsymbol{w}_t}(\boldsymbol{p}_t)$ be original and projected updates, respectively. Then, the effective update after the projection $\widehat{\boldsymbol{w}_{t+1}^p}$ lies on the geodesic on $\mathbb{S}^{d-1}$ defined by $\widehat{\boldsymbol{w}_t}$ and $\widehat{\boldsymbol{w}_{t+1}^o}$.*

Proof in Appendix. As such, we expect that our algorithm inherits the convergence guarantees of GD. As to the convergence rate, we conjecture that a similar analysis by Arora et al. (2019) holds.

The proposed method is readily adaptable to existing gradient-based optimization algorithms like SGD and Adam. Their modifications, SGDP and AdamP, are shown in Algorithms 1 and 2, respectively (Modifications are colorized). In practice, we consider two types of scale-invariance: layer-wise (*e.g.*, by LN) and channel-wise (*e.g.*, by BN, IN) invariance. Hence, the weight projection in equation 11 are performed either channel- or layer-wise. Our optimizers incur only 8% extra training time on top of the baselines (ResNet18 on ImageNet classification).

| **Algorithm 1: SGDP** |
| --- |
| **Require:** Learning rate $\eta > 0$, momentum $\beta > 0$, thresholds $\delta, \varepsilon > 0$. |
| 1: **while** $\boldsymbol{w}_t$ not converged **do** |
| 2:    $\boldsymbol{p}_t \leftarrow \beta \boldsymbol{p}_{t-1} + \nabla_{\boldsymbol{w}} f_t(\boldsymbol{w}_t)$ |
| 3:    Compute $\boldsymbol{q}_t$ with equation 11. |
| 4:    $\boldsymbol{w}_{t+1} \leftarrow \boldsymbol{w}_t - \eta \boldsymbol{q}_t$ |
| 5: **end while** |

| **Algorithm 2: AdamP** |
| --- |
| **Require:** Learning rate $\eta > 0$, momentum $0 < \beta_1, \beta_2 < 1$, thresholds $\delta, \varepsilon > 0$. |
| 1: **while** $\boldsymbol{w}_t$ not converged **do** |
| 2:    $\boldsymbol{m}_t \leftarrow \beta_1 \boldsymbol{m}_{t-1} + (1 - \beta_1) \nabla_{\boldsymbol{w}} f_t(\boldsymbol{w}_t)$ |
| 3:    $\boldsymbol{v}_t \leftarrow \beta_2 \boldsymbol{v}_{t-1} + (1 - \beta_2)(\nabla_{\boldsymbol{w}} f_t(\boldsymbol{w}_t))^2$ |
| 4:    $\boldsymbol{p}_t \leftarrow \boldsymbol{m}_t / (\sqrt{\boldsymbol{v}_t} + \varepsilon)$ |
| 5:    Compute $\boldsymbol{q}_t$ with equation 11. |
| 6:    $\boldsymbol{w}_{t+1} \leftarrow \boldsymbol{w}_t - \eta \boldsymbol{q}_t$ |
| 7: **end while** |

### 3.2 Empirical analysis of effective step sizes and the proposed projection

So far, we have studied the problem (§2), namely that the momentum accelerates effective step sizes for scale-invariant weights, as well as the corresponding solution (§3.1) only at a conceptual level. Here, we verify that the problem does indeed exist in practice and that our method successfully addresses it on both synthetic and real-world optimization problems.

**Synthetic simulation.** While examples of scale-invariant optimization problems abound in modern deep learning (*e.g.* BN), they are scant in popular toy optimization objectives designed for sanity checks. We use Rosenbrock function (Rosenbrock, 1960): $h(x_1, x_2) = (1 - x_1)^2 + 300(x_2 - x_1^2)^2$ (Figure 3). As our verification requires scale invariance, we define a 3D Rosenbrock function by adding a redundant radial axis $r$, while treating the original coordinates $(x_1, x_2)$ as the polar

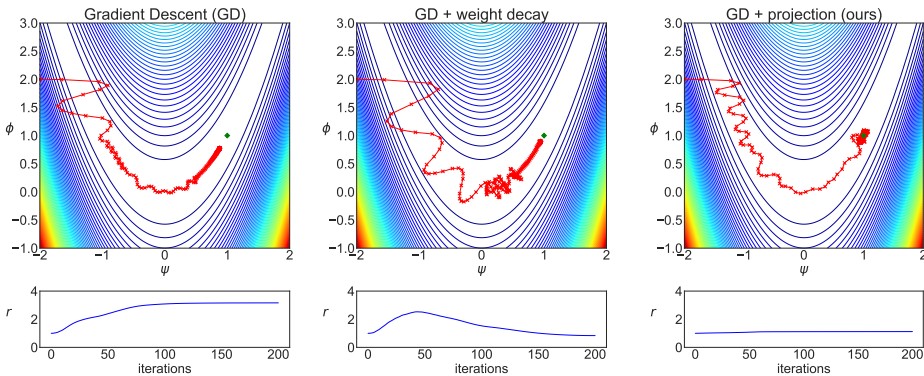

Figure 3. **3D scale-invariant Rosenbrock.** Three optimization algorithms are compared. Upper row: loss surface and optimization steps. Lower row: norm $r$ of parameters over the iterations. Results for Adam variants in Appendix §H.1.

angles $(\psi, \phi)$ of the spherical coordinates, resulting in the function $\widetilde{h}(r, \psi, \phi) = h(\psi, \phi)$. $\widetilde{h}$ is optimized in the 3D space with the Cartesian coordinates. We describe the full details in Appendix §B.

In Figure 3, we compare the trajectories for optimizers on the spherical coordinates $(\psi, \phi)$. We compare the baseline momentum GD and our projection solution. We additionally examine the impact of weight decay (WD), since a careful choice of WD is another way to regularize the norm growth. We observe that the momentum GD does not converge sufficiently to the optimum. The slowdown is explained by the decreased effective step sizes on $\mathbb{S}^2$ due to the increase in the parameter norm $(r = 1.0 \to 3.2)$. Careful tuning of WD partially addresses the slow convergence (momentum GD + WD trajectory) by regularizing the norm growth, but WD is still unable to preclude the initial surge in the weight norm $(r = 1.0 \to 2.53)$. In practice, addressing the problem with WD is even less attractive because WD is often a sensitive hyperparameter (see the following experiments). On the other hand, our projection solution successfully subdues the weight norm growth $(r = 1.0 \to 1.12)$, ensuring undiminished effective step sizes and a faster convergence to the optimum.

**Real-world experiments.** We verify the surge of weight norm and suboptimality of model performances in momentum-trained deep networks on real-world datasets: ImageNet classification with ResNet-18 and music tagging (Law et al., 2009) with Harmonic CNN (Won et al., 2020a). See Table 1 for the analysis. In all experiments, our projection solutions (SGDP, AdamP) restrain the weight norm growth much better than the vanilla momentum methods. For example, Adam-induced norm increase (+4.21) is 13.2 times greater than that of AdamP (+0.32) in the music tagging task. In ImageNet SGD experiments, we observe that a careful

Table 1. **Analysis of optimizers on real-world tasks.** The norm values and final performances for different tasks and optimizers are shown. Norm[1]: norm at first epoch. Norm[last]: norm at last epoch. Score: accuracy for ImageNet and AUC for music tagging.

| Task | Optimizer | WD | Norm[1] | Norm[last] | Score (↑) |
|---|---|---|---|---|---|
| ImageNet ResNet-18 | SGD | 0 | 1.35 | 4.34 (+2.99) | 67.94 |
| | SGD | $10^{-6}$ | 1.35 | 4.08 (+2.73) | 67.89 |
| | SGD | $10^{-5}$ | 1.33 | 2.70 (+1.36) | 69.06 |
| | SGD | $10^{-4}$ | 1.21 | 1.02 (-0.19) | 70.43 |
| | SGDP | 0 | 1.25 | 2.10 (+0.85) | 70.21 |
| | SGDP | $10^{-6}$ | 1.25 | 1.96 (+0.72) | 70.27 |
| | SGDP | $10^{-5}$ | 1.25 | 1.19 (-0.04) | **70.57** |
| | AdamW | 0 | 1.70 | 10.41 (+8.71) | 68.38 |
| | AdamP | 0 | 1.29 | 3.02 (+1.73) | **70.55** |
| Music tagging HCNN | AdamW | 0 | 7.35 | 11.56 (+4.21) | 90.86 |
| | AdamP | 0 | 7.30 | 7.62 (+0.32) | **91.19** |

choice of weight decay (WD) mitigates the norm increases, but the final norm values and performances are sensitive to the WD value. On the other hand, our SGDP results in stable final norm values and performances across different WD values, even at WD= 0. We observe that under the same learning setup models with smaller terminal weight norms tend to obtain improved performances. Though it is difficult to elicit a causal relationship, we verify in §4 that SGDP and AdamP bring about performance gains in a diverse set of real-world tasks. More analysis around the norm growth and the learning curves are in Appendix §F. We also conduct analysis on the momentum coefficient in Appendix §H.3

## 4 EXPERIMENTS

In this section, we demonstrate the effectiveness of our projection module for training scale-invariant weights with momentum-based optimizers. We experiment over various real-world tasks and datasets. From the image domain, we show results on ImageNet classification (§4.1, §D.2, §D.3), object detection (§4.2), and robustness benchmarks (§4.3, §D.1). From the audio domain, we study music tagging, speech recognition, and sound event detection (§4.4). We further show the results when the scale invariance is artificially introduced to a network with no scale-invariant parameters (*e.g.* Transformer-XL (Dai et al., 2019)) in §4.5. To diversify the root cause of scale invariances, we consider the case where it stems from the $\ell_2$ projection of the features, as opposed to the statistical normalization done in *e.g.* BN, in the image retrieval experiments (§4.6). In the above set of experiments totaling $> 10$ setups, our proposed modifications (SGDP and AdamP) bring about consistent performance gains against the baselines (SGD (Sutskever et al., 2013) and AdamW (Loshchilov & Hutter, 2019)). We provide the implementation details in the Appendix §E and the standard deviation values for the experiments in Appendix §H.2.

### 4.1 IMAGE CLASSIFICATION

Batch normalization (BN) and momentum-based optimizer are standard techniques to train state-of-the-art image classification models (He et al., 2016; Han et al., 2017; Sandler et al., 2018; Tan & Le, 2019; Han et al., 2020). We evaluate the proposed method with ResNet (He et al., 2016), one of the most popular and powerful architectures on ImageNet, and MobileNetV2 (Sandler et al., 2018), a relatively lightweight model with ReLU6 and depthwise convolutions, on the ImageNet-1K benchmark (Russakovsky et al., 2015). For ResNet, we employ the training hyperparameters in (He et al., 2016). For MobileNetV2, we have searched for the best hyperparameters, as it is generally difficult to train it with the usual settings. Recent researches have identified better training setups (Cubuk et al., 2020) where the cosine-annealed learning rates and larger training epochs (100 epochs or 150 epochs) than 90 epochs (He et al., 2016) are used. We use those setups for all experiments in this subsection.

Our optimizers are compared against their corresponding baselines in Table 2. Note that AdamP is compared against AdamW (Loshchilov & Hutter, 2019), which has closed the gap between Adam and SGD performances on large-scale benchmarks. Across the spectrum of network sizes, our optimizers outperform the baselines. Even when the state-of-the-art CutMix (Yun et al., 2019) regularization is applied, our optimizers introduce further gains. We provide three additional experiments on EfficientNet (Tan & Le, 2019) (§D.2), the large-batch training scenario (§D.3) and the comparison in the same computation cost (§H.4) to demonstrate the benefit of AdamP on diverse training setups. There, again, our methods outperform the baselines.

Table 2. **ImageNet classification.** Accuracies of state-of-the-art networks trained with SGDP and AdamP.

| Architecture | # params | SGD | SGDP (ours) | Adam | AdamW | AdamP (ours) |
|---|---|---|---|---|---|---|
| MobileNetV2 | 3.5M | 71.55 | **72.09** (+0.54) | 69.32 | 71.21 | **72.45** (+1.24) |
| ResNet18 | 11.7M | 70.47 | **70.70** (+0.23) | 68.05 | 70.39 | **70.82** (+0.43) |
| ResNet50 | 25.6M | 76.57 | **76.66** (+0.09) | 71.87 | 76.54 | **76.92** (+0.38) |
| ResNet50 + CutMix | 25.6M | 77.69 | **77.77** (+0.08) | 76.35 | 78.04 | **78.22** (+0.18) |

### 4.2 OBJECT DETECTION

Object detection is another widely-used real-world task where the models often include normalization layers and are trained with momentum-based optimizers. We study the two detectors Center-Net (Zhou et al., 2019) and SSD (Liu et al., 2016a) to verify that the proposed optimizers are also applicable to various objective functions beyond the classification task. The detectors are either initialized with the ImageNet-pretrained network (official PyTorch models) or trained from scratch, in order to separate the effect of our method from that

Table 3. **MS-COCO object detection.** Average precision (AP) scores of CenterNet (Zhou et al., 2019) and SSD (Liu et al., 2016a) trained with Adam and AdamP optimizers.

| Model | Initialize | Adam | AdamP (ours) |
|---|---|---|---|
| CenterNet | Random | 26.57 | **27.11** (+0.54) |
| CenterNet | ImageNet | 28.29 | **29.05** (+0.76) |
| SSD | Random | 27.10 | **27.97** (+0.87) |
| SSD | ImageNet | 28.39 | **28.67** (+0.28) |

of the pretraining. ResNet18 (He et al., 2016) and VGG16 BN (Simonyan & Zisserman, 2015) are used for the CenterNet and SSD backbones, respectively. In Table 3, we report average precision performances based on the MS-COCO (Lin et al., 2014) evaluation protocol. We observe that AdamP boosts the performance against the baselines. It demonstrates the versatility of our optimizers.

## 4.3 ROBUSTNESS

Model robustness is an emerging problem in the real-world applications and the training of robust models often involve complex optimization problems (*e.g.* minimax). We examine how our optimizers stabilize the complex optimization. We consider two types of robustness: adversarial (below) and cross-bias (Bahng et al., 2020) (Appendix §D.1).

Adversarial training alternatively optimizes a minimax problem where the inner optimization is an adversarial attack and the outer optimization is the standard classification problem. Adam is commonly employed, in order to handle the complexity of the optimization (Tsipras et al., 2019; Chun et al., 2019).

We consider solving the minimax optimization problem with our proposed optimizers. We train Wide-ResNet (Zagoruyko & Komodakis, 2016) with the projected gradient descent (PGD) (Madry et al., 2018) on CIFAR-10. We use 10 inner PGD iterations and $\varepsilon = 80/255$ for the $L_2$ PGD and $\varepsilon = 4/255$ for the $L_\infty$ PGD. Figure 4 shows the learning curves of

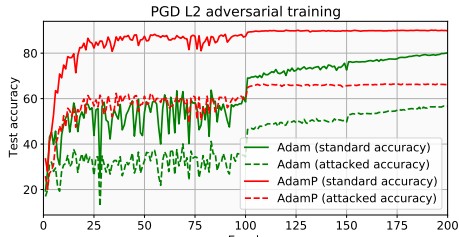

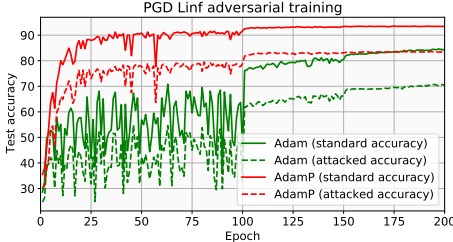

Figure 4. **Adversarial training.** Learning curves by Adam and AdamP.

Adam and AdamP. By handling the effective step sizes, AdamP achieves a faster convergence than Adam (less than half the epochs required). AdamP brings more than $+9.3$ pp performance gap in all settings. We also performed $\varepsilon = 8/255$ and the results are reported in Appendix §E.4.1.

## 4.4 AUDIO CLASSIFICATION

We evaluate the proposed optimizer on three audio classification tasks with different physical properties: music clips, verbal audios, and acoustic signals. For automatic music tagging, we use the MagnaTagATune (MTAT) benchmark (Law et al., 2009) with 21k samples and 50 tags. Each clip contains multiple tags. We use the Speech Commands dataset (Warden, 2018) for the keyword spotting task (106k samples, 35 classes, single label). For acoustic signals, we use the DCASE sound event detection benchmark (Mesaros et al., 2017) (53k samples, 17 tags, multi-labeled).

We train the Harmonic CNN (Won et al., 2020a) on the three benchmarks. Harmonic CNN consists of data-driven harmonic filters and stacked convolutional filters with BN. Audio datasets are usually smaller than the image datasets and are multi-labeled, posing another set of challenges for the optimization. Won et al. (2020a) have trained the network with a mixture of Adam and SGD (Won et al., 2019b). Instead of the mixture solution, we have searched the best hyperparameters for the Adam and AdamP on a validation set. The results are given in Table 4. AdamP shows better performances than the baselines, without having to adopt the complex mixture solution. The results signify the superiority of AdamP for training scale-invariant weights on the audio domain.

Table 4. **Audio classification.** Results on three audio tasks with Harmonic CNN (Won et al., 2020a).

| Optimizer | Music Tagging | | Keyword Spotting | Sound Event Tagging |
|---|---|---|---|---|
| | ROC-AUC | PR-AUC | Accuracy | F1 score |
| Adam + SGD (Won et al., 2019b) | 91.27 | 45.67 | 96.08 | 54.60 |
| AdamW | 91.12 | 45.61 | 96.47 | 55.24 |
| AdamP (ours) | **91.35** (+0.23) | **45.79** (+0.18) | **96.89** (+0.42) | **56.04** (+0.80) |

### 4.5 LANGUAGE MODELING

Models in language domains, such as the Transformer (Vaswani et al., 2017), often do not have any scale-invariant weight. The use of layer normalization (LN) in the Transformer does not result in the scale invariance because LN is applied right after the skip connection $f(\boldsymbol{w}, \boldsymbol{x}) := \text{LN}(g_{\boldsymbol{w}}(\boldsymbol{x}) + \boldsymbol{x})$ (note the difference from equation 2). The skip

Table 5. **Language Modeling.** Perplexity on Wiki-Text103. Lower is better.

| Model | AdamW | AdamP (ours) |
|---|---|---|
| Transformer-XL | 23.38 | **23.26** (-0.12) |
| Transformer-XL + WN | 23.96 | **22.77** (-1.19) |

connection makes $f$ scale-*variant* with respect to $\boldsymbol{w}$. To allow our optimizers to effectively operate on Transformer, we introduce scale invariance artificially through the weight normalization (WN) (Salimans & Kingma, 2016).

We have trained Transformer-XL (Dai et al., 2019) on WikiText-103 (Merity et al., 2016). As shown in Table 5, AdamP does not significantly outperform AdamW (23.33→23.26) without the explicit enforcement of scale invariance. WN on its own harms the performance (23.33→23.90 for Adam). When AdamP is applied on Transformer-XL + WN, it significantly improves over the AdamW baseline (23.90→22.73), beating the original performance 23.33 by Adam on the vanilla network.

### 4.6 RETRIEVAL

In the previous experiments, we have examined the scale invariances induced by statistical normalization methods (*e.g.* BN). Here, we consider another source of scale invariance, $\ell_2$ projection of features, which induces a scale invariance in the preceding weights. It is widely used in retrieval tasks for more efficient distance computations and better performances. We fine-tune the ImageNet-pretrained ResNet-50 network on CUB (Wah et al., 2011), Cars-196 (Krause et al., 2013), In-Shop Clothes (Liu et al., 2016b), and Stanford Online Products (SOP) (Oh Song et al., 2016) benchmarks with the triplet (Schroff et al., 2015) and the ProxyAnchor (Kim et al., 2020) losses. In Table 6, we observe that AdamP outperforms Adam over all four image retrieval datasets. The results support the superiority of AdamP for networks with $\ell_2$ normalized embeddings.

Table 6. **Image retrieval.** Optimizers are tested on networks with $\ell_2$-induced scale invariant weights. Recall@1.

| Optimizer | CUB | | Cars-196 | | InShop | | SOP | |
|---|---|---|---|---|---|---|---|---|
| | Triplet | PA | Triplet | PA | Triplet | PA | Triplet | PA |
| AdamW | 57.9 | 69.3 | 59.8 | 86.7 | 62.7 | 85.2 | 62.0 | 76.5 |
| AdamP (ours) | **58.2** (+0.3) | **69.5** (+0.2) | **59.9** (+0.2) | **86.9** (+0.2) | **62.8** (+0.0) | **87.4** (+2.2) | **62.6** (+0.6) | **78.0** (+1.5) |

## 5 CONCLUSION

Momentum-based optimizers induce an excessive growth of the scale-invariant weight norms. The growth of weight norms prematurely decays the effective optimization steps, leading to sub-optimal performances. The phenomenon is prevalent in many commonly-used setup. Momentum-based optimizers (*e.g.* SGD and Adam) are used for training the vast majority of deep models. The widespread use of normalization layers make a large proportion of network weights scale-invariant (*e.g.* ResNet). We propose a simple and effective solution: project out the radial component from the optimization updates. The resulting SGDP and AdamP successfully suppress the weight norm growth and train a model at an unobstructed speed. Empirically, our optimizers have demonstrated their superiority over the baselines on more than 10 real-world learning tasks.

### ACKNOWLEDGEMENT

We thank NAVER AI LAB colleagues for discussion and advice, especially Junsuk Choe for the internal review. Naver Smart Machine Learning (NSML) platform (Kim et al., 2018) has been used in the experiments.

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

## Appendix

This document provides additional materials for the main paper. Content includes the proofs (§A), detailed experimental setups (§B and §E), and the additional analysis on the learning rate scheduling and weight decay (§F).

## A    PROOFS FOR THE CLAIMS

We provide proofs for Lemma 2.2, Corollary 2.3, and Proposition 3.1 in the main paper.

**Lemma A.1** (Monotonic norm growth by the momentum). *For a scale-invariant parameter $\boldsymbol{w}$ updated via equation 7, we have*

$$\|\boldsymbol{w}_{t+1}\|_2^2 = \|\boldsymbol{w}_t\|_2^2 + \eta^2 \|\boldsymbol{p}_t\|_2^2 + 2\eta^2 \sum_{k=0}^{t-1} \beta^{t-k} \|\boldsymbol{p}_k\|_2^2. \tag{A.1}$$

*Proof.* From equation 7, we have

$$\|\boldsymbol{w}_{t+1}\|_2^2 = \|\boldsymbol{w}_t\|_2^2 + \eta^2 \|\boldsymbol{p}_t\|_2^2 - 2\eta \boldsymbol{w}_t \cdot \boldsymbol{p}_t \tag{A.2}$$

It remains to prove that $\boldsymbol{w}_t \cdot \boldsymbol{p}_t = -\eta \sum_{k=0}^{t-1} \beta^{t-k} \|\boldsymbol{p}_k\|_2^2$. We prove by induction on $t \geq 0$.

First, when $t = 0$, we have $\boldsymbol{w}_0 \cdot \boldsymbol{p}_0 = \boldsymbol{w}_0 \cdot \nabla_{\boldsymbol{w}} f(\boldsymbol{w}_0) = 0$ because of equation 4.

Now, assuming that $\boldsymbol{w}_\tau \cdot \boldsymbol{p}_\tau = -\eta \sum_{k=0}^{\tau-1} \beta^{\tau-k} \|\boldsymbol{p}_k\|_2^2$, we have

$$\boldsymbol{w}_{\tau+1} \cdot \boldsymbol{p}_{\tau+1} = \boldsymbol{w}_{\tau+1} \cdot (\beta \boldsymbol{p}_\tau + \nabla_{\boldsymbol{w}} f(\boldsymbol{w}_{\tau+1})) = \beta \boldsymbol{w}_{\tau+1} \cdot \boldsymbol{p}_\tau = \beta(\boldsymbol{w}_\tau - \eta \boldsymbol{p}_\tau) \cdot \boldsymbol{p}_\tau \tag{A.3}$$

$$= -\beta\eta \sum_{k=0}^{\tau-1} \beta^{\tau-k} \|\boldsymbol{p}_k\|_2^2 - \beta\eta \|\boldsymbol{p}_\tau\|_2^2 = -\eta \sum_{k=0}^{\tau} \beta^{\tau-k+1} \|\boldsymbol{p}_k\|_2^2 \tag{A.4}$$

which completes the proof.  □

**Corollary A.2** (Asymptotic norm growth comparison). *Let $\|\boldsymbol{w}_t^{\mathrm{GD}}\|_2$ and $\|\boldsymbol{w}_t^{\mathrm{GDM}}\|_2$ be the weight norms at step $t \geq 0$, following the vanilla gradient descent growth (Lemma 2.1) and momentum-based gradient descent growth (Lemma 2.2), respectively. We assume that the norms of the updates $\|\boldsymbol{p}_t\|_2$ for GD with and without momentum are identical for every $t \geq 0$. We further assume that the sum of the update norms is non-zero and bounded: $0 < \sum_{t \geq 0} \|\boldsymbol{p}_t\|_2^2 < \infty$. Then, the asymptotic ratio between the two norms is given by:*

$$\frac{\|\boldsymbol{w}_t^{\mathrm{GDM}}\|_2^2 - \|\boldsymbol{w}_0\|_2^2}{\|\boldsymbol{w}_t^{\mathrm{GD}}\|_2^2 - \|\boldsymbol{w}_0\|_2^2} \longrightarrow 1 + \frac{2\beta}{1-\beta} \quad \text{as} \quad t \to \infty. \tag{A.5}$$

*Proof.* From Lemma 2.1 and Lemma 2.2, we obtain

$$\|\boldsymbol{w}_t^{\mathrm{GD}}\|_2^2 - \|\boldsymbol{w}_0\|_2^2 = \eta^2 \sum_{k=0}^{t-1} \|\boldsymbol{p}_k\|_2^2 \tag{A.6}$$

$$\|\boldsymbol{w}_t^{\mathrm{GDM}}\|_2^2 - \|\boldsymbol{w}_0\|_2^2 = \eta^2 \sum_{k=0}^{t-1} \|\boldsymbol{p}_k\|_2^2 + 2\eta^2 \sum_{k=0}^{t-1} \left( \sum_{l=1}^{t-1-k} \beta^l \right) \|\boldsymbol{p}_k\|_2^2. \tag{A.7}$$

Thus, the corollary boils down to the claim that

$$F_t := \frac{\sum_{k=0}^{t} \left( \sum_{l=1}^{t-k} \beta^l \right) A_k}{\sum_{k=0}^{t} A_k} \longrightarrow \frac{\beta}{1-\beta} \quad \text{as} \quad t \to \infty \tag{A.8}$$

where $A_k := \|\boldsymbol{p}_k\|_2^2$.

Let $\epsilon > 0$. We will find a large-enough $t$ that bounds $F_t$ around $\frac{\beta}{1-\beta}$ by a constant multiple of $\epsilon$.

We first let $T$ be large enough such that

$$\sum_{k \geq T+1} A_k \leq \epsilon \tag{A.9}$$

which is possible because $\sum_{t \geq 0} A_t < \infty$. We then let $T'$ be large enough such that

$$\frac{\beta}{1-\beta} - \sum_{l=1}^{T'} \beta^l \leq \frac{\epsilon}{T \max_k A_k} \tag{A.10}$$

which is possible due to the convergence of the geometric sum and the boundedness of $A_k$ (because its infinite sum is bounded).

We then define $t = T + T'$ and break down the sums in $F_t$ as follows:

$$F_t = \frac{\sum_{k=0}^{T} \left( \sum_{l=1}^{T+T'-k} \beta^l \right) A_k + \sum_{k=T+1}^{T+T'} \left( \sum_{l=1}^{T+T'-k} \beta^l \right) A_k}{\sum_{k=0}^{T} A_k + \sum_{k=T+1}^{T+T'} A_k} \tag{A.11}$$

$$= \frac{\sum_{k=0}^{T} \left( \frac{\beta}{1-\beta} + r_1(\epsilon) \right) A_k + r_2(\epsilon)}{\sum_{k=0}^{T} A_k + r_3(\epsilon)} \tag{A.12}$$

$$\leq \frac{\frac{\beta}{1-\beta} \sum_{k=0}^{T} A_k + T \max_k A_k r_1(\epsilon) + r_2(\epsilon)}{\sum_{k=0}^{T} A_k + r_3(\epsilon)} \tag{A.13}$$

where $r_1$, $r_2$, and $r_3$ are the residual terms that are bounded as follows:

$$|r_1(\epsilon)| \leq \frac{\epsilon}{T \max_k A_k} \tag{A.14}$$

by equation A.10 and

$$|r_2(\epsilon)| \leq \frac{(1-\beta)\epsilon}{\beta} \quad \text{and} \quad |r_3(\epsilon)| \leq \epsilon \tag{A.15}$$

by equation A.9.

It follows that

$$\left| F_t - \frac{\beta}{1-\beta} \right| \leq \left| \frac{-\frac{\beta}{1-\beta} r_3(\epsilon) + T \max_k A_k r_1(\epsilon) + r_2(\epsilon)}{\sum_{k=0}^{T} A_k + r_3(\epsilon)} \right| \tag{A.16}$$

$$\leq \frac{1}{\sum_{k=0}^{T} A_k} \left( \frac{\beta}{1-\beta} + \frac{1-\beta}{\beta} + 1 \right) \epsilon \tag{A.17}$$

$$\leq \frac{1}{M} \left( \frac{\beta}{1-\beta} + \frac{1-\beta}{\beta} + 1 \right) \epsilon \tag{A.18}$$

due to the triangular inequality and the positivity of $r_3$. $M > 0$ is a suitable constant independent of $T$. $\qquad\square$

**Proposition A.3** (Effective update direction after projection). *Let $\boldsymbol{w}_{t+1}^{\text{o}} := \boldsymbol{w}_t - \eta \boldsymbol{p}_t$ and $\boldsymbol{w}_{t+1}^{\text{p}} := \boldsymbol{w}_t - \eta \Pi_{\boldsymbol{w}_t}(\boldsymbol{p}_t)$ be original and projected updates, respectively. Then, the effective update after the projection $\widehat{\boldsymbol{w}_{t+1}^{\text{p}}}$ lies on the geodesic on $\mathbb{S}^{d-1}$ defined by $\widehat{\boldsymbol{w}_t}$ and $\widehat{\boldsymbol{w}_{t+1}^{\text{o}}}$.*

*Proof.* The geodesic defined by $\widehat{\boldsymbol{w}_t}$ and $\widehat{\boldsymbol{w}_{t+1}^{\text{o}}}$ can be written as $\mathbb{S}^{d-1} \cap \text{span}(\boldsymbol{w}_t, \boldsymbol{p}_t)$. Thus, it suffices to show that $\widehat{\boldsymbol{w}_{t+1}^{\text{p}}} \in \text{span}(\boldsymbol{w}_t, \boldsymbol{p}_t)$. Indeed, we observe that $\widehat{\boldsymbol{w}_{t+1}^{\text{p}}} := \frac{\boldsymbol{w}_t - \eta \Pi_{\boldsymbol{w}_t}(\boldsymbol{p}_t)}{\|\boldsymbol{w}_t - \eta \Pi_{\boldsymbol{w}_t}(\boldsymbol{p}_t)\|_2} \in \text{span}(\boldsymbol{w}_t, \boldsymbol{p}_t)$ because $\Pi_{\boldsymbol{w}_t}(\boldsymbol{p}_t) = \boldsymbol{p}_t - (\widehat{\boldsymbol{w}_t} \cdot \boldsymbol{p}_t)\widehat{\boldsymbol{w}_t}$. $\qquad\square$

## B  TOY EXAMPLE DETAILS

**2D toy example in Figure 1**  We describe the details of the toy example in Figure 1. We solve the following optimization problem:

$$\min_{\boldsymbol{w}} \frac{\boldsymbol{w}}{\|\boldsymbol{w}\|_2} \cdot \frac{\boldsymbol{w}^\star}{\|\boldsymbol{w}^\star\|_2} \tag{B.1}$$

where $\boldsymbol{w}$ and $\boldsymbol{w}^\star$ are 2-dimensional vectors. The problem is identical to the maximization of the cosine similarity between $\boldsymbol{w}$ and $\boldsymbol{w}^\star$. We set the $\boldsymbol{w}^\star$ to $(0, -1)$ and the initial $\boldsymbol{w}$ to $(0.001, 1)$.

This toy example has two interesting properties. First, the normalization term makes the optimal $\boldsymbol{w}$ for the problem not unique: if $\boldsymbol{w}^\star$ is optimal, then $c\boldsymbol{w}^\star$ is optimal for any $c > 0$. In fact, the cost function is scale-invariant. Second, the cost function is not convex.

As demonstrated in Figure 1 and videos attached in our submitted code, the momentum gradient method fails to optimize equation B.1 because of the excessive norm increases. In particular, our simulation results show that a larger momentum induces a larger norm increase (maximum norm 2.93 when momentum is 0.9, and 27.87 when momentum is 0.99), as we shown in the main paper § 2.4. On the other hand, our method converges most quickly, among the compared methods, by taking advantage of the momentum-induced accelerated convergence, while avoiding the excessive norm increase.

**3D spherical toy example in Figure 3**  We employ 3D Rosenbrock function (Rosenbrock, 1960):
$\widetilde{h}(r, \psi, \phi) = (1 - \psi)^2 + 300(\phi - \psi^2)^2$ in the spherical coordinate $(r, \psi, \phi)$. Since we are mapping 2D space to a spherical surface, $(\psi, \phi)$ is defined on the hemisphere. Thus, $-\pi/2 \leq (\psi, \phi) \leq \pi/2$. Based on the required range of problem $f$, scalar $c$ can be multiplied to each angles $f(c\psi, c\phi)$ to adjust angle scale to problem scale, where we choose $c = 1.5$ in the experiments. The initial point is $(c\psi, c\phi) = (-2, 2)$ above the unit sphere $r = 1$, and the minimum point is $(c\psi, c\phi) = (1, 1)$.

Instead of optimizing $h$ in the spherical coordinate, we optimize the toy example in the Cartesian coordinate $(x, y, z)$ by computing $\min_{x,y,z} \widetilde{h}(r, \psi, \phi) = \min_{x,y,z} \widetilde{h}(T(x, y, z))$. We employ the spherical transform $T : (x, y, z) \rightarrow (r, \psi, \phi)$ as follows:

$$r = \sqrt{x^2 + y^2 + z^2}, \quad \psi = \cos^{-1}(x/z), \quad \phi = \sin^{-1}(y/r). \tag{B.2}$$

For all optimizers, we set the momentum to 0.9, and we have exhaustively searched the optimal initial learning rates between 0.001 and 0.1. The learning rates are decayed by linearly at every iteration.

## C  $\delta$ SENSITIVITY

Based on ImageNet training with ResNet18, we analyzed the cosine similarity of scale-invariant parameter and scale-variant parameter, and verified the sensitivity of $\delta$. We first measured the cosine similarity between the gradient and weights (Eq. 11). As a results, the cosine similarities for scale-variant weights are $[0.0028, 0.5660]$ , compared to $[5.5 \times 10^{-10}, 4.4 \times 10^{-6}]$ for scale-invariant ones (99% confidence intervals). Because of this large gap, our methods are stable on a wide range of $\delta$ values. We also measured scale-variant and scale-invariant parameter detection accuracy based on Eq. 11 with various $\delta$ values. The

Table B.1. $\delta$ **sensitivity.** We measured scale-variant and scale-invariant parameter detection performance of AdamP. AdamP consistently shows high performance over a wide range of $\delta$ values.

| $\delta$ | Scale-variant detection | Scale-invariant detection | Accuracy |
|---|---|---|---|
| 1 | 86.64% | 100% | 70.74 |
| 0.2 | 100% | 100% | 70.83 |
| 0.1 | 100% | 100% | 70.81 |
| 0.05 | 100% | 100% | 70.78 |
| 0.02 | 100% | 99.94% | 70.81 |
| 0.01 | 100% | 97.73% | 70.66 |

results are shown in Table B.1. In a wide range of delta values, AdamP perfectly discriminates the scale-variant and scale-invariant parameters. Also, AdamP consistently shows high performance. Therefore, AdamP is not sensitive to the $\delta$ value and $\delta = 0.1$ is suitable to separate scale-variant and scale-invariant parameters.

# D  ADDITIONAL EXPERIMENTS

## D.1  CROSS-BIAS ROBUSTNESS

Cross-bias generalization problem (Bahng et al., 2020) tackles the scenario where the training and test distributions have different real-world biases. This often occurs when the training-set biases provide an easy shortcut to solve the problem. Bahng et al. (2020) has proposed the ReBias scheme based on the minimax optimization, where the inner problem maximizes the independence between an intentionally biased representation and the target model of interest and the outer optimization solves the standard classification problem. As for the adversarial training, Bahng et al. (2020) has employed the Adam to handle the complex optimization.

We follow the two cross-bias generalization benchmarks proposed by Bahng et al. (2020). The first benchmark is the Biased MNIST, the dataset synthesized by injecting colors on the MNIST background pixels. Each sample is colored according to a pre-defined class-color mapping with probability $\rho$. The color is selected at random with $1 - \rho$ chance. For example, $\rho = 1.0$ leads a completely biased dataset and $\rho = 0.1$ leads to an unbiased dataset. Each model is trained on the $\rho$-biased MNIST and tested on the unbiased MNIST. We train a stacked convolutional network with BN and ReLU. The second benchmark is the 9-Class ImageNet representing the real-world biases, such as textures (Geirhos et al., 2019). The unbiased accuracy is measured by pseudo-labels generated by the texture clustering. We also report the performance on ImageNet-A (Hendrycks et al., 2019), the collection of failure samples of existing CNNs.

In Table D.1, we observe that AdamP outperforms Adam in all the benchmarks. AdamP is a good alternative to Adam for difficult optimization problems applied on scale-invariant parameters.

Table D.1. **Real-world bias robustness.** Biased MNIST and 9-Class ImageNet benchmarks with Re-Bias (Bahng et al., 2020).

| Optimizer | Biased MNIST Unbiased acc. at $\rho$ | | | | | 9-Class ImageNet | | |
|---|---|---|---|---|---|---|---|---|
| | .999 | .997 | .995 | .990 | avg. | Biased | UnBiased | ImageNet-A |
| Adam | 22.9 | 63.0 | 74.9 | 87.0 | 61.9 | 93.8 | 92.6 | 31.2 |
| AdamP (ours) | 30.5 (+7.5) | 70.9 (+7.9) | 80.9 (+6.0) | 89.6 (+2.6) | 68.0 (+6.0) | 95.2 (+1.4) | 94.5 (+1.8) | 32.9 (+1.7) |

## D.2  TRAINING WITH VARIOUS TECHNIQUES

EfficientNet (Tan & Le, 2019) and ReXNet (Han et al., 2020) are recently proposed high-performance networks that is trained using various techniques such as data augmentation (Cubuk et al., 2018; 2020), stochastic depth (Huang et al., 2016), and dropout (Srivastava et al., 2014). We measured the performance of AdamP on EfficientNets and ReXNets to verify it can be used with other training techniques. The experiment were conducted on the well-known image classification codebase[1]. Table D.2 shows performance of the original paper, our reproduced results, and AdamP. The results show that AdamP is still effective in training with various techniques, and can contribute to improving the best performance of the network.

Table D.2. **Training with various techniques.** EfficientNet and ReXNet were trained using the latest training techniques, and the result shows that AdamP can also contribute to learning with these techniques.

| Network | Image size | # of params | Paper | Reproduce | AdamP |
|---|---|---|---|---|---|
| EfficientNet-B0 | 224 | 5.3M | 77.1 | 77.7 | 78.1 (+0.4) |
| EfficientNet-B1 | 240 | 7.8M | 79.1 | 78.7 | 79.9 (+1.2) |
| EfficientNet-B2 | 260 | 9.1M | 80.1 | 80.4 | 80.5 (+0.1) |
| EfficientNet-B3 | 300 | 12.2M | 81.6 | 81.5 | 81.9 (+0.4) |
| ReXNet-×1.0 | 224 | 4.8M | 77.9 | 77.9 | 78.1 (+0.2) |
| ReXNet-×1.3 | 224 | 6.4M | 79.5 | 79.5 | 79.7 (+0.2) |

---

[1] https://github.com/rwightman/pytorch-image-models

Table E.1. **Dataset statistics.** Summary of the dataset specs used in the experiments. ImageNet-1k (Russakovsky et al., 2015), MS-COCO (Lin et al., 2014), CIFAR-10 (Krizhevsky, 2009), Biased-MNIST (LeCun et al., 1998; Bahng et al., 2020), ImageNet-A (Hendrycks et al., 2019), 9-Class ImageNet (Bahng et al., 2020), 9-Class ImageNet-A (Bahng et al., 2020), MagnaTagATune (Law et al., 2009), Speech Commands (Warden, 2018), DCASE 2017 task 4 (Mesaros et al., 2017), CUB (Wah et al., 2011), In-Shop Clothes (Liu et al., 2016b) and SOP (Oh Song et al., 2016) are used in experiments.

| Task | Dataset | #classes | #samples | Note |
|---|---|---|---|---|
| Image classification | ImageNet-1k | 1,000 | $\approx 1.33M$ | |
| Object detection | MS-COCO | 80 | $\approx 123k$ | |
| Robustness | CIFAR-10 | 10 | $\approx 60k$ | |
| | Biased-MNIST | 10 | $\approx 60k$ | colors are injected to be biased |
| | 9-Class ImageNet | 9 | $\approx 57k$ | a subset of ImageNet-1k |
| | 9-Class ImageNet-A | 9 | 617 | a subset of ImageNet-A |
| Audio classification | MagnaTagATune | 50 | $\approx 21k$ | mutl-labeled dataset |
| | Speech Commands | 35 | $\approx 106k$ | |
| | DCASE 2017 task 4 | 17 | $\approx 53k$ | mutl-labeled dataset |
| Language Modeling | WikiText-103 | - | $\approx 103M$ tokens | vocabulary size (267,735) |
| Image retrieval | CUB | 200 | $\approx 12k$ | tr classes (100), te classes (100) |
| | Cars-196 | 196 | $\approx 16k$ | tr classes (98), te classes (98) |
| | In-Shop Clothes | 7,982 | $\approx 53k$ | tr classes (3,997), te classes (3985) |
| | SOP | 22,634 | $\approx 120k$ | tr classes (11,318), te classes (11,316) |

## D.3 LARGE-BATCH TRAINING

In order to efficiently use multiple machines and huge computational resources, large-batch training is essential. However, a general optimizer suffers from a significant performance decrease in a large-batch training, so large-batch training is another challenge for a deep learning optimizer. (You et al., 2017; 2019; Goyal et al., 2017) We conducted experiments to verify the performance of AdamP in such large-batch training, and the results are shown in Table D.3. The performance improvement of AdamP in large-batch training is greater than that of regular batch-size (Table 2). Therefore, the decrease of effective learning rate due to momentum can be considered as one of the causes of performance degradation in large-batch training. However, AdamP does not show as much performance as the large-batch optimizers (You et al., 2017; 2019; Goyal et al., 2017), and therefore, applying AdamP to the large-batch optimizer should be studied as a future work.

Table D.3. **Large-batch training.** ResNet50 (He et al., 2016) is trained over 100 epochs with various batch-size on ImageNet (Russakovsky et al., 2015). AdamW (Loshchilov & Hutter, 2019) and AdamP are used as the optimizer.

| Batch-size | AdamW | AdamP (ours) |
|---|---|---|
| 4k | 72.41 | **73.75** (+1.34) |
| 8k | 69.74 | **71.36** (+1.62) |
| 16k | 63.79 | **66.89** (+3.1) |

## E EXPERIMENTS SETTINGS

We describe the experimental settings in full detail for reproducibility.

## E.1 COMMON SETTINGS

All experiments are conducted based on PyTorch. SGDP and AdamP are implemented to handle channel-wise (*e.g.* batch normalization (Ioffe & Szegedy, 2015) and instance normalization (Ulyanov et al., 2016)) and layer-wise normalization (Ba et al., 2016). Based on the empirical measurement of the inner product between the weight vector and the corresponding gradient vector for scale-invariant parameters (they are supposed to be orthogonal), we set the $\delta$ in Algorithms 1 and 2 to 0.1. We use the decoupled weight decay (Loshchilov & Hutter, 2019) for SGDP and AdamP in order to separate the gradient due to the weight decay from the gradient due to the loss function. Please refer to the attached codes: `sgdp.py` and `adamp.py` for further details.

## E.2    Image classification

Experiments on ResNet (He et al., 2016) are conducted based on the standard settings : learning rate 0.1, weight decay $10^{-4}$, batch-size 256, momentum 0.9 with Nesterov (Sutskever et al., 2013) for SGD and SGDP. For Adam series, we use the learning rate 0.001, weight decay $10^{-4}$, batch-size 256, $\beta_1$ 0.9, $\beta_2$ 0.999, $\epsilon$ $10^{-8}$.

For training MobileNetV2 (Sandler et al., 2018), we have additionally used label-smoothing and large batch size 1024, and have searched the best learning rates and weight decay values for each optimizer.

The training sessions are run for 100 epochs (ResNet18, ResNet50) or 150 epochs (MobileNetV2, ResNet50 + CutMix) with the cosine learning rate schedule (Loshchilov & Hutter, 2016) on a machine with four NVIDIA V100 GPUs.

## E.3    Object detection

Object detection performances have been measured on the MS-COCO dataset (Lin et al., 2014) with two popular object detectors: CenterNet (Zhou et al., 2019) and SSD (Liu et al., 2016a). We adopt the CenterNet with ResNet18 (He et al., 2016) backbone and the SSD with VGG16 BN (Simonyan & Zisserman, 2015) backbone as baseline detectors. CenterNet has been trained for 140 epochs with learning rate $2.5 \times 10^{-4}$, weight decay $10^{-5}$, batch size 64, and the cosine learning rate schedule. SSD has been trained for 110 epochs with learning rate $10^{-4}$, weight decay $10^{-5}$, batch size 64, and the step learning rate schedule which decays learning rates by $1/10$ at $70\%$ and $90\%$ of training.

## E.4    Robustness

### E.4.1    Adversarial training

Adversarial robustness benchmark results have been reproduced using the unofficial PyTorch implementation of the adversarial training of Wide-ResNet (Zagoruyko & Komodakis, 2016)[2] for the CIFAR-10 attack challenge[3]. Projected gradient descent (PGD) attack variants (Madry et al., 2018) have been used as the threat model for the all the experiments. We employed 10 inner PGD iterations and $\varepsilon = 80/255$ for the $L_2$ PGD attack and $\varepsilon = 4/255$ for the $L_\infty$ PGD attack. We additionally test stronger threat models, $L_\infty$ PGD with $\varepsilon = 8/255$ and 20 iterations as Madry et al. (2018). Following Madry et al. (2018), we employ 7 inner PGD iterations for the training threat model, and 20 inner PGD iterations for the test threat model. In all the experiments, Wide-ResNet-34-10 have been trained with the PGD threat model. The models have been trained for 200 epochs with learning rate 0.01, weight decay 0.0002, batch size 128, and the step learning rate schedule which decays learning rates by $1/10$ at epochs 100 and 150. Table E.2 shows the detailed results.

Table E.2. **Adversarial training.** Standard and attacked accuracies of PGD-adversarially trained Wide-ResNet on CIFAR-10. For each threat model scenario, we report the perturbation size $\varepsilon$ and the number of PGD iterations $n$. $*$ denotes results from the original paper.

| Attack Method (train) | Attack Method (test) | Optimizer | Standard Acc | Attacked Acc |
|---|---|---|---|---|
| $\ell_\infty$ ($\varepsilon = 4/255, n = 10$) | $\ell_\infty$ ($\varepsilon = 4/255, n = 10$) | Adam
AdamP | 80.12
**89.85 (+9.73)** | 56.58
**66.28 (+9.70)** |
| | $\ell_\infty$ ($\varepsilon = 8/255, n = 20$) | Adam
AdamP | 80.12
**89.85 (+9.73)** | 29.00
**35.76 (+6.76)** |
| $\ell_2$ ($\varepsilon = 80/255, n = 10$) | $\ell_2$ ($\varepsilon = 80/255, n = 10$) | Adam
AdamP | 84.14
**93.46 (+9.32)** | 70.33
**83.59 (+13.26)** |
| $\ell_\infty$ ($\varepsilon = 8/255, n = 7$) | $\ell_\infty$ ($\varepsilon = 8/255, n = 20$) | Adam
AdamP | 69.76
**85.74 (+15.98)** | 39.48
**42.42 (+2.94)** |
| | | Madry et al. (2018) | - | 45.8* |

---

[2] https://github.com/louis2889184/pytorch-adversarial-training
[3] https://github.com/MadryLab/cifar10_challenge

### E.4.2 ROBUSTNESS AGAINST REAL-WORLD BIASES

We follow the two cross-bias generalization benchmarks proposed by (Bahng et al., 2020). We refer (Bahng et al., 2020) for interested readers. For all experiments, the batch size is 256 and 128 for Biased MNIST and 9-Class ImageNet, respectively. For Biased MNIST, the initial learning rate is 0.001, decayed by factor 0.1 every 20 epochs. For 9-Class ImageNet, the learning rate is 0.001, decayed by cosine annealing. We train the fully convolutional network and ResNet18 for 80 and 120 epochs, respectively. The weight decay is $10^{-4}$ for all experiments.

### E.5 AUDIO CLASSIFICATION

**Dataset.**    Three datasets with different physical properties are employed as the audio benchmarks. We illustrate the statistics in Table E.1. The **music tagging** is a multi-label classification task for the prediction of user-generated tags, *e.g.*, genres, moods, and instruments. We use a subset of MagnaTagATune (MTAT) dataset (Law et al., 2009) which contains ≈21k audio clips and 50 tags. The average of tag-wise Area Under Receiver Operating Characteristic Curve (ROC-AUC) and Area Under Precision-Recall Curve (PR-AUC) are used as the evaluation metrics. **Keyword spotting** is a primitive speech recognition task where an audio clip containing a keyword is categorized among a list of limited vocabulary. We use the Speech Commands dataset (Warden, 2018) which contains ≈ 106k samples and 35 command classes such as "yes", "no", "left", "right". The accuracy metric is used for the evaluation. **Acoustic sound detection** is a multi-label classification task with non-music and non-verbal audios. We use the "large-scale weakly supervised sound event detection for smart cars" dataset used for the DCASE 2017 challenge (Mesaros et al., 2017). It has ≈53k audio clips with 17 events such as "Car", "Fire truck", and "Train horn". For evaluation, we use the F1-score by setting the prediction threshold as 0.1.

**Training setting.**    We use the 16kHz sampling rate for the all experiments, and all hyperparameters, *e.g.*, the number of harmonics, trainable parameters, are set to the same as in (Won et al., 2020a). The official implementation by (Won et al., 2020a)[4] is used for all the experiments. We compare three different optimizers, Adam, AdamP (ours), and the complex mixture of Adam and SGD proposed by (Won et al., 2019b).

For the mixture of Adam and SGD, we adopt the same hyperparameters as in the previous papers (Won et al., 2019b;c;a; 2020a). The mixed optimization algorithm first runs Adam for 60 epochs with learning rate $10^{-4}$. After 60 epochs, the model with the best validation performance is selected as the initialization for the second phase. During the second phase, the model is trained using SGD for 140 epochs with the learning rate $10^{-4}$, decayed by 1/10 at epochs 20 and 40. We use the weight decay $10^{-4}$ for the optimizers. Using the hyperparameters, we reproduce the ROC-AUC score on MTAT dataset by the recent clean-up paper Won et al. (2020b), 91.27.

To show the effectiveness of our method, we have searched the best hyperparameters for the Adam optimizer on the MTAT validation dataset and have transferred them to AdamP experiments. As the result of our search, we set the weight decay as 0 and the initial learning rate as 0.0001 decayed by the cosine annealing scheduler. The number of training epochs are set to 100 for MTAT dataset and 30 for SpeechCommand and DCASE dataset. As a result, we observe that AdamP shows superior performances compared to the complex mixture, with a fewer number of training epochs (200 → 30).

### E.6 RETRIEVAL

**Dataset.**    We use four retrieval benchmark datasets. For the CUB (Wah et al., 2011) dataset which contains bird images with 200 classes, we use 100 classes for training and the rest for evaluation. For evaluation, we query every test image to the test dataset, and measure the recall@1 metric. The same protocol is applied to Cars-196 (Krause et al., 2013) (196 classes) and SOP (Oh Song et al., 2016) (22,634 classes) datasets. For InShop (Liu et al., 2016b) experiments, we follow the official benchmark setting proposed by (Liu et al., 2016b). We summarize the dataset statistics in Table E.1

---

[4]https://github.com/minzwon/data-driven-harmonic-filters

**Training setting.** For the all experiments, we use the same backbone network and the same training setting excepting the optimizer and the loss function. The official implementation by (Kim et al., 2020)[5] is used for the all experiments.

We use the Pytorch official ImageNet-pretrained ResNet50 model as the initialization. During the training, we freeze the BN statistics as the ImageNet statistics (`eval` mode in PyTorch). We replace the global average pooling (GAP) layer of ResNet with the summation of GAP and global max pooling layer as in the implementation provided by (Kim et al., 2020). Pooled features are linearly mapped to the 512 dimension embedding space and $\ell_2$-normalized.

We set the initial learning rate $10^{-4}$, decayed by the factor $0.5$ for every 5 epochs. Every mini-batch contains 120 randomly chosen samples. For the better stability, we train only the last linear layer for the first 5 epochs, and update all the parameters for the remaining steps. The weight decay is set to 0.

## F   ANALYSIS WITH LEARNING RATE SCHEDULE AND WEIGHT DECAY

We analyze the norm growth of scale-invariant parameters and the corresponding change in effective step-size. We provide extended results of this experiment by measuring norm growth and effective step-size for SGD, SGDP, Adam and AdamP under various weight decay values. The experiment is based on ResNet18 trained on the ImageNet dataset, and the network was trained for 100 epoch in the standard setting as in E.2. We have analyzed the impact of learning rate schedule and weight decay for the scale-invariant parameters. Figure F.1 and Figure F.2 show the results of SGD and SGDP under the step-decay and cosine-annealing learning rate schedules, respectively. The same results for Adam and AdamP are shown in Figures F.3 and Figure F.4. We have used the optimal weight decay value of the baseline as the reference point and changed the weight decay values. We write the weight decay in each experiment in relative values with respect to the corresponding optimal values.

In all considered settings, SGDP and AdamP effectively prevent the norm growth, which prevents the rapid decrease of the effective step sizes. SGDP and AdamP shows better performances than the baselines. Another way to prevent the norm growth is to control the weight decay. However, this way of norm adjustment is sensitive to the weight decay value and results in poor performances as soon as non-optimal weight decay values are used. Figure F.1 and F.3 shows that the learning curves are generally sensitive to the weight decay values even showing abnormalities such as the gradual increase of the effective step sizes. On the other hand, SGDP and AdamP prevent rapid norm growth without weight decay, leading to smooth effective step size reduction. SGDP and AdamP are not sensitive to the weight decay values.

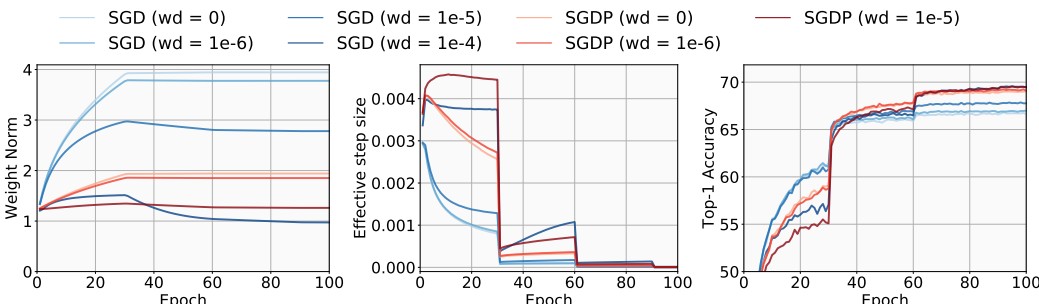

Figure F.1. Norm value analysis: SGD + step learning rate decay.

### F.1   ANALYSIS AT HIGH WEIGHT DECAY

In the previous Figures, we only showed the case where the weight decay is less than 1e-4. This is because when the weight decay is large, the scale of the graph changes and it is difficult to demonstrate the difference in small weight decay. Therefore, we separately report the large weight decay cases through Figure F.5 and F.6. The high weight decay further reduces the weight norm and increases

---

[5] https://github.com/tjddus9597/Proxy-Anchor-CVPR2020

the effective step-size, but it does not lead to an improvement in performance. Also, this result shows that the weight decay used in Section 4.1 (1e-4) is the best value for the baseline optimizers.

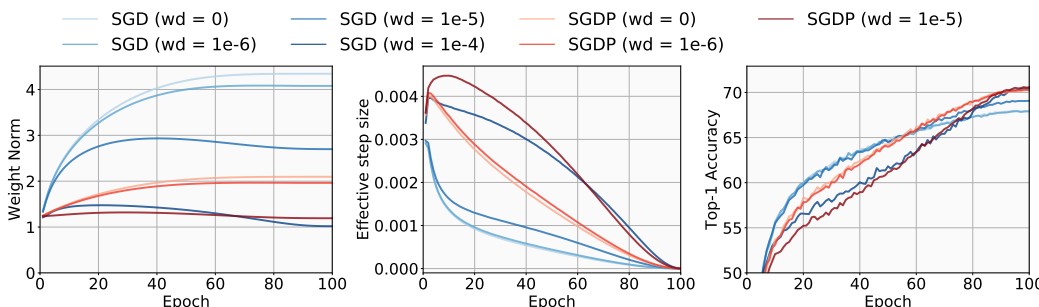

Figure F.2. Norm value analysis: SGD + cosine learning rate decay. SGD (wd=$10^{-4}$) and SGDP (wd=$10^{-5}$) are the same setting as the reported numbers in Table 2

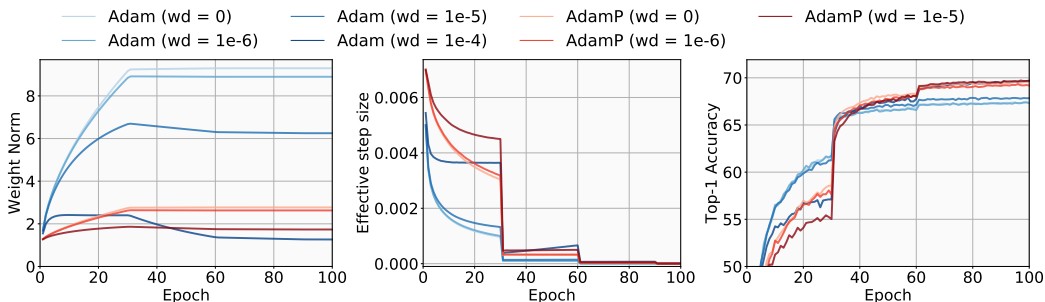

Figure F.3. Norm value analysis: AdamW + step learning rate decay.

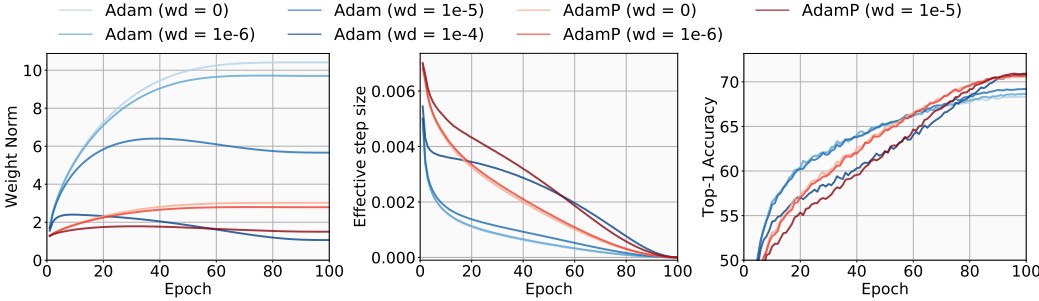

Figure F.4. Norm value analysis: AdamW + cosine learning rate decay. AdamW (wd=$10^{-4}$) and AdamP (wd=$10^{-6}$) are the same setting as the reported numbers in Table 2

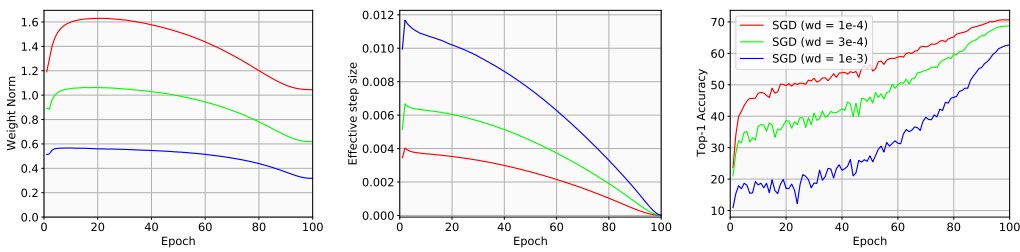

Figure F.5. Norm value analysis for high weight decay: SGD + cosine learning rate decay.

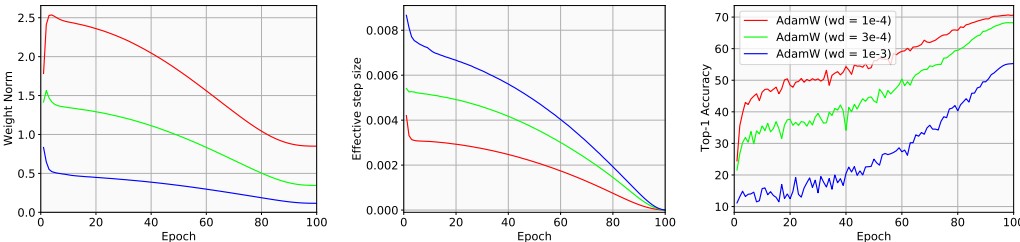

Figure F.6. Norm value analysis for high weight decay: AdamW + cosine learning rate decay.

## G  RELATED WORK

We provide a brief overview of related prior work. A line of work is dedicated to the development general and effective optimizers, such as Adagrad (Duchi et al., 2011), Adam (Kingma & Ba, 2015), and RMSprop. Researchers have sought strategies to improve Adam through *e.g.* improved convergence (Reddi et al., 2018), warmup learning rate (Liu et al., 2020), moving average (Zhang et al., 2019b), Nesterov momentum (Dozat, 2016), rectified weight decay (Loshchilov & Hutter, 2019), and variance of gradients (Zhuang et al., 2020). Another line of researches studies existing optimization algorithms in greater depth. For example, (Hoffer et al., 2018; Arora et al., 2019; Zhang et al., 2019a) have delved into the effective learning rates on scale-invariant weights. This paper at the intersection between the two. We study the issues when momentum-based optimizers are applied on scale-invariant weights. We then propose a new optimization method to address the problem.

### G.1  COMPARISON WITH CHO & LEE (2017)

Cho & Lee (2017) have proposed optimizers that are similarly motivated from the scale invariance of certain parameters in a neural network. They have also propose a solution that reduces the radial component of the optimization steps. Despite the apparent similarities, the crucial difference between Cho & Lee (2017) and ours is in the space where the optimization is performed. Cho & Lee (2017) performs the gradient steps on the Riemannian manifold. Ours project the updates on the tangent planes of the manifold. Thus, ours operates on the same Euclidean space where SGD and Adam operate on. From a theory point of view, Cho & Lee (2017) has made contributions to the optimization theory on a Riemannian manifold. Our contributions are along a different axis: we focus on the norm growth when the updates are projected onto the tangent spaces (§2.4, §3.1, and §3.2). We contribute present theoretical findings that are not covered by Cho & Lee (2017).

From the practicality point of view, we note that changing the very nature of space from Euclidean to Riemannian requires users to find the sensible ranges for many optimization hyperparameters again. For example, Cho & Lee (2017) has "used different learning rates for the weights in Euclidean space and on Grassmann [Riemannian] manifolds" (page 7 of (Cho & Lee, 2017)), while in our case hyperparameters are largely compatible between scale-invariant and scale-variant parameters, for they are both accommodated in the same kind of space. We have shown that SGDP and AdamP outperform the SGD and Adam baselines with exactly the same optimization hyperparameters (Section E.2). The widely used Xavier (Glorot & Bengio, 2010) or Gaussian initializations are no longer available in the spherical Riemannian manifold, necessitating changes in the code defining parameters and their initialization schemes: *e.g.* Cho & Lee (2017) has employed a dedicated initialization based on truncated normals. Finally, Cho & Lee (2017) requires users to manually register scale-invariant hyperparameters. This procedure is not scalable, as the networks nowadays are becoming deeper and more complex, and the architectures are becoming more machine-designed than handcrafted. Our optimizers automatically detect scale invariances through the orthogonality test (Equation 11), and users do not need to register anything by themselves. The shift from linear to curved coordinates introduces non-trivial changes in the optimization settings; our optimizers does not introduce such a shift and it is much easier to apply our method on a new kind of model, from a user's perspective.

In addition to the above conceptual considerations, we compare the performances between ours and the Grassmann optimizers (Cho & Lee, 2017). We first compare them in the 3D scale-invariant Rosenbrock example (see §3.2 for a description). Figure G.1 and G.2 show the results for the Grass-

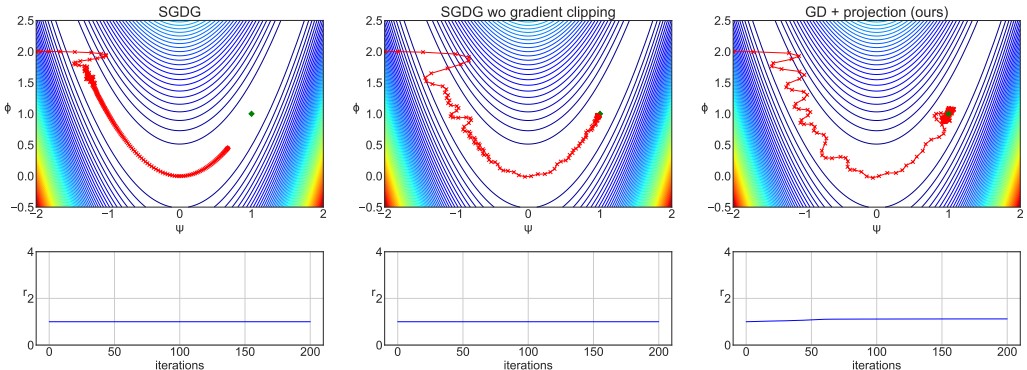

Figure G.1. **3D scale-invariant Rosenbrock with SGDG optimizers.** 3D toy experiments based on SGDG optimizers. Upper row: loss surface and optimization steps. Lower row: norm $r$ of parameters over the iterations.

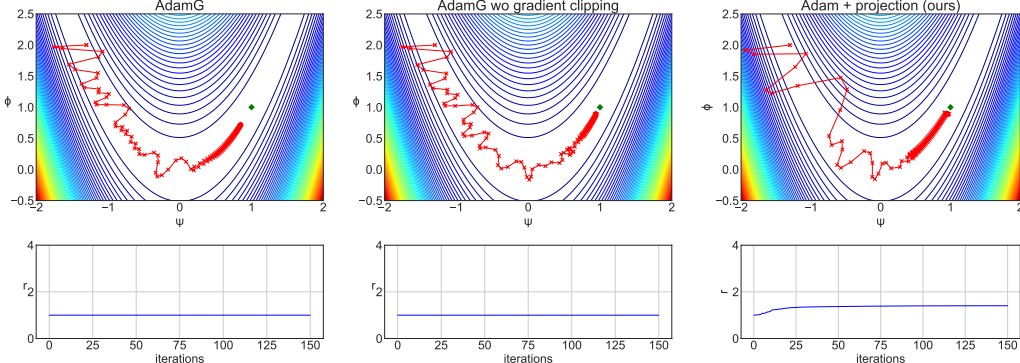

Figure G.2. **3D scale-invariant Rosenbrock with AdamG optimizers.** 3D toy experiments based on AdamG optimizers. Upper row: loss surface and optimization steps. Lower row: norm $r$ of parameters over the iterations.

mann optimizers: SGDG and AdamG, respectively. It can be seen that SGDG and AdamG optimizer do not introduce any norm increase by definition: they operate on a spherical space. However, rate of convergence seems slower than our SGDP and AdamP (the rightmost plots in each figure). We note that SGDG and AdamG include a gradient clipping operation to restrict the magnitude of projected gradients on the Grassmann manifold to $\nu$, where $\nu$ is empirically set to 0.1 in Cho & Lee (2017). We have identified an adverse effect of the gradient clipping, at least on our toy example. As the clipping is removed, SGDG and AdamG come closer to the fast convergence of ours (the middle plots in each figure). We may conclude for the toy example that the Grassmann optimizers also address the unnecessary norm increase and converge as quickly as our optimizers do.

For a more practical setup, we present experiments on ImageNet with the Grassmann optimizers (SGDG and AdamG) and report the top-1 accuracies. We have conducted the experiments with ResNet18 following the settings in Table 2. In addition to the learning rate of optimizers ($\mathrm{lr}_{\text{Euclidean}}$), Cho & Lee (2017) introduces three more hyperparameters: (1) learning rate on the Grassmann manifold ($\mathrm{lr}_{\text{Grassmann}}$), (2) degree of the regularization on the Grassmann manifold ($\alpha$), which replaces the $L_2$ regularization in Euclidean space, and (3) the gradient clipping threshold ($\nu$). Since Cho & Lee (2017) have not reported results on ImageNet training, we have tuned the hyperparameters ourselves, following the guidelines of Cho & Lee (2017). Table G.1 shows the exploration of the hyperparameters that are considered above. The first rows show the performance with the recommended hyperparameters in the paper ($\alpha = \nu = 0.1$ and $\mathrm{lr}_{\text{Euclidean}} = 0.01$ for SGDG and AdamG; $\mathrm{lr}_{\text{Grassmann}} = 0.2$ for SGDG and $\mathrm{lr}_{\text{Grassmann}} = 0.05$ for AdamG). We first tested the effects of regularization ($\alpha$) and gradient clipping ($\nu$), which were additionally introduced in the Grassmann optimizers. The first block of the Table G.1 shows the result. The gradient clipping ($\nu$) did not have a significant effect, but the regularization ($\alpha$) decrease the performance. Therefore, we turned off regularization ($\alpha = 0$) in the optimizers and started learning rate search. In Table 2, the baseline optimizers and our optimizers are compared in the fixed learning rate (SGD: 0.1, Adam: 0.001)

Table G.1. **Hyperparameter search of the Grassmann optimizers.** This talbe shows the performance of ResNet18 in ImageNet trained with Grassmann optimizers (SGDG and AdamG) with various hyperparameters. Except for the hyperparameters specified in the table, the setting is the same as the experiment in Table 2.

(a) SGDG (SGDP accuracy : **70.70**)

| Search target | $\alpha$ | $\nu$ | lr Euclidean | lr Grassmann | Accuracy |
|---|---|---|---|---|---|
| $\alpha,\nu$ | 0.1 | 0.1 | 0.01 | 0.2 | 66.00 |
| | 0 | 0.1 | 0.01 | 0.2 | **67.85** |
| | 0.1 | None | 0.01 | 0.2 | 66.35 |
| lr | 0 | 0.1 | 0.01 | 0.2 | 67.85 |
| | 0 | 0.1 | 0.1 | 2 | 63.28 |
| | 0 | 0.1 | 0.005 | 0.1 | 68.73 |
| | 0 | 0.1 | 0.1 | 0.1 | **69.74** |
| lr tuning | 0 | 0.1 | 0.1 | 0.1 | 69.74 |
| | 0 | 0.1 | 0.3 | 0.3 | 68.33 |
| | 0 | 0.1 | 0.03 | 0.03 | **70.39** |
| | 0 | 0.1 | 0.01 | 0.01 | 69.21 |

(b) AdamG (AdamP accuracy : **70.82**)

| Search target | $\alpha$ | $\nu$ | lr Euclidean | lr Grassmann | Accuracy |
|---|---|---|---|---|---|
| $\alpha,\nu$ | 0.1 | 0.1 | 0.01 | 0.05 | 67.40 |
| | 0 | 0.1 | 0.01 | 0.05 | **68.43** |
| | 0.1 | None | 0.01 | 0.05 | 67.36 |
| lr | 0 | 0.1 | 0.01 | 0.05 | **68.43** |
| | 0 | 0.1 | 0.001 | 0.005 | 66.97 |
| | 0 | 0.1 | 0.0002 | 0.001 | 55.9 |
| | 0 | 0.1 | 0.001 | 0.001 | 59.36 |
| lr tuning | 0 | 0.1 | 0.01 | 0.05 | 68.43 |
| | 0 | 0.1 | 0.03 | 0.15 | 63.96 |
| | 0 | 0.1 | 0.003 | 0.015 | **69.45** |
| | 0 | 0.1 | 0.1 | 0.5 | 59.39 |

setting. So, for fair comparison, we design four learning rate (lr) candidates for grassmann optimizers: 1) Use the learning rates of the paper (Cho & Lee, 2017). 2) & 3) Use the fixed learning rates for $lr_{euclidean}$ or $lr_{grassmann}$, and adjust the other following the learning rate ratio of the paper. 4) Use baseline learning rates for both learning rates ($lr_{euclidean}$ and $lr_{grassmann}$). The result is reported in the second block of Table G.1. SGDG shows the best performance at option 4) and AdamG is the best in option 1). However, the performances of Grassmann optimizers are still much lower than our optimizers: SGDP (70.70) and AdamP (70.82). We further tuned the learning rate of the Grassmann optimizer, which is shown in the last block of the Table G.1. After learning rate tuning, Grassmann optimizers show comparable performance with the baseline optimizer (SGD: 70.47, Adam: 68.05, AdamW: 70.39). However, learning rate tuning is essential for the performance, and our optimizer has a higher performance. So, it can be said that our optimizer is more practical and effective for ImageNet training than the Grassmann optimizer.

## H ADDITIONAL EXPERIMENTS

### H.1 3D TOY EXPERIMENTS FOR ADAM

We also evaluated our 3D toy experiments for Adam optimizer. The results are shown in Fig. H.1 Adam optimizer shows quiet different steps in early stage. However, the fact that norm growth reduces the rate of late convergence is the same as SGD. The weight decay and our projection mitigate the norm growth and helps fast convergence.

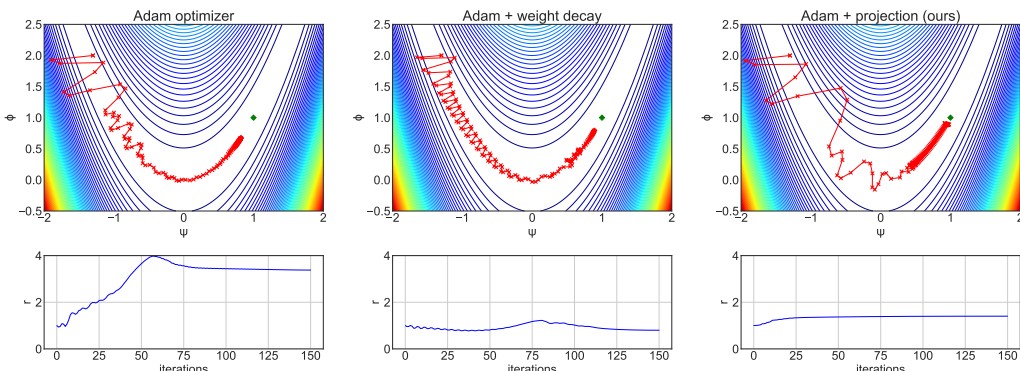

Figure H.1. **3D scale-invariant Rosenbrock with Adam optimizers.** 3D toy experiments based on Adam optimizers. Upper row: loss surface and optimization steps. Lower row: norm $r$ of parameters over the iterations.

## H.2 STANDARD DEVIATION OF EXPERIMENTS

The most experiments in the paper were performed three times with different seed. The mean value was reported in the main paper and the standard deviation value is shown in the Table H.1, H.2, H.3, H.4, H.5 accordingly. In case of ImageNet classification, mean values are shown in Table 2 and standard deviation values are shown in Table H.1. In most cases, the improvement of our optimizer is significantly larger than the standard deviation. Even in the worse case, (SGDP on ResNet50), the performance increases are at the level of the standard deviations. For the audio classification results (Table 4 and H.2), the performance increases by AdamP in Music Tagging (ROC-AUC) and Keyword Spotting are much greater than the standard deviation values; for the other entries, the performance increases are at the level of the standard deviations. In the language model, all experiments have similar standard deviation values as shown in Table H.3. In all cases, AdamP's improvement (Table 5) is significantly larger than the standard deviation. We observe a different tendency for the image retrieval tasks (Table 6 and H.4). In many cases, the standard deviation values are large, so the performance boost is not clearly attributable to AdamP. However, the performance increases for InShop-PA and SOP-PA are still greater than the standard deviation values. Finally, in the results for robustness against real-world biases (Table D.1 and H.5), our optimization algorithm brings about far greater performance gains than the randomness among trials. In most experiments, the performance improvement of our optimizers is much greater than the standard deviation values.

Table H.1. **Standard deviation for ImageNet classification.** Standard deviation of accuracy of state-of-the-art networks trained with SGDP and AdamP (Table 2).

| Architecture | # params | SGD | SGDP (ours) | Adam | AdamW | AdamP (ours) |
|---|---|---|---|---|---|---|
| MobileNetV2 | 3.5M | ± 0.14 | ± 0.11 | ± 0.04 | ± 0.16 | ± 0.16 |
| ResNet18 | 11.7M | ± 0.16 | ± 0.06 | ± 0.06 | ± 0.07 | ± 0.06 |
| ResNet50 | 25.6M | ± 0.06 | ± 0.08 | ± 0.12 | ± 0.06 | ± 0.02 |
| ResNet50 + CutMix | 25.6M | ± 0.11 | ± 0.05 | ± 0.08 | ± 0.09 | ± 0.09 |

Table H.2. **Standard deviation for audio classification.** Standard deviation for results on the audio tasks with Harmonic CNN (Won et al., 2020a) (Table 4).

| Optimizer | Music Tagging | | Keyword Spotting | Sound Event Tagging |
|---|---|---|---|---|
| | ROC-AUC | PR-AUC | Accuracy | F1 score |
| Adam + SGD (Won et al., 2019b) | ± 0.06 | ± 0.03 | - | - |
| AdamW | ± 0.08 | ± 0.19 | ± 0.06 | ± 0.39 |
| AdamP (ours) | ± 0.07 | ± 0.27 | ± 0.06 | ± 0.83 |

Table H.3. **Standard deviation for Language Modeling.** Standard deviation of the perplexity values on Wiki-Text103 (Table 5).

| Model | AdamW | AdamP (ours) |
|---|---|---|
| Transformer-XL | ± 0.05 | ± 0.05 |
| Transformer-XL + WN | ± 0.06 | ± 0.06 |

Table H.4. **Standard deviation for image retrieval.** Standard deviation of Recall@1 on the retrieval tasks (Table 6).

| Optimizer | CUB | | Cars-196 | | InShop | | SOP | |
|---|---|---|---|---|---|---|---|---|
| | Triplet | PA | Triplet | PA | Triplet | PA | Triplet | PA |
| AdamW | ± 0.91 | ± 0.24 | ± 0.82 | ± 0.09 | ± 0.52 | ± 0.08 | ± 0.71 | ± 0.04 |
| AdamP (ours) | ± 0.62 | ± 0.77 | ± 1.35 | ± 0.19 | ± 0.82 | ± 0.02 | ± 0.74 | ± 0.09 |

## H.3 ANALYSIS WITH MOMENTUM COEFFICIENT

Since our optimizer is deeply involved with the momentum of optimizers, we measure and analyze the effect of our optimizer on several momentum coefficients. The experiment was conducted using ResNet18 in ImageNet with the setting of Section 4.1, and weight decay was not used to exclude norm decrease due to weight decay. The results are shown in Table H.6. According to the difference between equation 8 and equation 12, the effect of preventing norm growth of our optimizer is

Table H.5. **Standard deviation for robustness against real-world biases.** Standard deviation for Re-Bias (Bahng et al., 2020) performances on Biased MNIST and 9-Class ImageNet benchmarks (Table D.1).

| Optimizer | Biased MNIST Unbiased acc. at $\rho$ | | | | | 9-Class ImageNet | | |
|---|---|---|---|---|---|---|---|---|
| | .999 | .997 | .995 | .990 | avg. | Biased | UnBiased | ImageNet-A |
| Adam | ± 6.04 | ± 0.48 | ± 0.70 | ± 0.48 | - | ± 0.27 | ± 0.40 | ±1.06 |
| AdamP (ours) | ± 1.73 | ± 1.31 | ± 2.16 | ± 0.40 | - | ± 0.21 | ± 0.27 | ± 0.82 |

affected by the momentum coefficient. It can be observed in this experiment. The larger the momentum coefficient, the greater the norm difference between SGD and SGDP. In addition, it can be seen that the improvement in accuracy of SGDP also increases as the momentum coefficient increases. The experiment shows that our optimizer can be used with most of the momentum coefficients, especially when the momentum coefficient is large, the effect of our optimizer is significant and essential for the high performance.

Table H.6. **Analysis with momentum coefficient.** We measured the difference between our SGDP and SGD with various momentum coefficients.

| Momentum | SGD | | SGDP | | Difference | |
|---|---|---|---|---|---|---|
| | $\text{Norm}^{\text{last}}$ | Accuracy | $\text{Norm}^{\text{last}}$ | Accuracy | $\text{Norm}^{\text{last}}$ | Accuracy |
| 0.5 | 7.60 | 67.57 | 6.88 | 67.75 | -0.7 | 0.18 |
| 0.8 | 8.31 | 67.64 | 6.57 | 68.55 | -1.7 | 0.91 |
| 0.9 | 8.53 | 67.42 | 6.33 | 68.95 | -2.2 | 1.53 |
| 0.95 | 8.76 | 67.67 | 6.23 | 69.12 | -2.5 | 1.45 |
| 0.975 | 8.73 | 67.56 | 6.14 | 69.17 | -2.6 | 1.60 |

## H.4 COMPARISON AT THE SAME COMPUTATION COST

As specified in Section 3.1, our optimizers requires an additional computation cost, which increases the training cost by 8%. In general, the optimizer's performance is compared in the same iteration, and we followed this convention in other experiments. However, the training cost is also an important issue, so we conduct further verification of our optimizer through comparison at the same training budget. The experimental setup is simple. We performed imagenet classification in Section 4.1 with only 92% epochs for our optimizers (SGDP and AdamP) and set the training budget of our optimizer and baseline optimizer to be the same. The results are shown in Table H.7. Training iteration is reduced, so the performance of our optimizer is reduced, but it still outperforms the baseline optimizer. Thus, it can be seen that our optimizer outperforms the baseline optimizer not only on the same iteration, but also on the same training budget.

Table H.7. **ImageNet classification comparison at the same computation cost.** Accuracies of state-of-the-art networks trained with SGDP and AdamP. We also conduct training over 92% epochs with SGDP and AdamP for the comparison in the same computation cost.

| Architecture | SGD | SGDP (92%) | SGDP | AdamW | AdamP (92%) | AdamP |
|---|---|---|---|---|---|---|
| MobileNetV2 | 71.55 | 71.85 (+0.30) | 72.09 (+0.54) | 71.21 | 72.34 (+0.13) | 72.45 (+1.24) |
| ResNet18 | 70.47 | 70.54 (+0.07) | 70.70 (+0.23) | 70.39 | 70.64 (+0.25) | 70.82 (+0.43) |
| ResNet50 | 76.57 | 76.58 (+0.01) | 76.66 (+0.09) | 76.54 | 76.84 (+0.30) | 76.92 (+0.38) |

