# OpenReview forum: "AdamP: Slowing Down the Slowdown for Momentum Optimizers on Scale-invariant Weights"
_ICLR.cc/2021/Conference — ICLR 2021 Poster_

### Official Review · AnonReviewer4 · 2020-10-23
**Nice paper that adresses the negative impact of momentum on scale-invariant parameters, but misses comparison with [1]**

**Rating:** 7
**Confidence:** 4

**Review:**

### Summary:
This paper studies the hurtful effect of momentum on scale-invariant parameters (such as weight matrices of linear layers followed by Batch Normalization (BN)): Indeed, momentum tend to increase the norm of the parameters, but the effective update size of scale-invariant parameters is inversely proportional to their norm, which leads to "_premature decay of effective step size_". The authors propose the improved SGDP and AdamP to reduce this issue, and show how it improves the performances of a wide variety of models (ResNets, MobileNets, Transformers, etc...) on many different tasks (ImageNet classification, adversarial training, audio classification, etc...)

### Strengths:
+ This paper addresses the optimization of scale-invariant parameters, which can be found in pretty much all the state-of-the-art models.
+ The paper is well written and contains a good balance of illustrative examples, theoretical analysis and experiments. Good job!
+ The empirical evaluation of the proposed algorithms is quite large, and many tasks and architectures are considered.
+ I also really like the "cos(w, Grad_w)" hack to automatically detect scale invariant parameters in the model. This is a very neat trick!

### Concern:
My main concern is with the statement that "_our paper is first to delve into the issue in the widely-used combination of scale-invariant parameters and momentum-based optimizers"_ . Indeed the authors have missed [1], which also adapts SGD with momentum and Adam to work on scale-invariant parameters, and showed performance improvement over vanilla SGD and Adam. Now, the approach of [1] is slightly different, in the sense that they propose to keep the scale-invariant parameters on the unit sphere S^1. Also, [1] performs the projection onto the tangent space before computing the momentum term (and second order moment in Adam), while this projection is done after in SGDP and AdamP.

In any case, I do believe proper comparison with [1] throughout the paper is required, due to the conceptual similarity of both methods. It would be particularly relevant  to show how the angles between SGDP and the SGD of [1] compare in Figure 1 (I think they would be identical, although not 100% certain). Also, I'm curious to see if the proposed SGDP and AdamP work better than the algorithms proposed in [1], as the nice automatic learning rate decay of BN disappears when keeping the scale-invariant parameters on the unit sphere (which would be an advantage of the proposed SGDP and AdamP).

### Reasons for score:
Overall I really liked that paper, but I don't think it would be fair to accept it without an in-depth comparison with [1].

### Questions / Comments:
- I like the toy example in Figure 1, although I'm not sure I understand what you mean by: "_Compared to GD, GD+Momentum speeds up the norm growth, resulting in a slower effective convergence in S^1, though being faster in the nominal space R^2_". It seems to me that the angle between the optimal solution and GD+momentum is smaller than the angle between the optimal solution and GD. I'm not sure to understand how do you observe a slower effective convergence in  S^1?
- (3.1, real-world experiments and F) What happens if you increase WD when using SGD? It seems that the best values reported for SGD in Table 1 and Figures F.* are with WD=1e-4, which is also the highest value reported. One can imagine even better results could be obtained using WD=3e-4 or 1e-3. It would be nice to have this result for completeness.
- Reporting mean and standard deviation across several seeds would have been nice, but I'm going to let this one slide, since results are reported on a lot of different tasks and architectures.

[1] Cho, Minhyung and Lee, Jaehyung, _Riemannian approach to batch normalization_, NIPS 2017

___

### After author response and paper revision

First, I would like to congratulate the authors for their amazing work during the rebuttal period.

My main concern, the comparison against [1], has been perfectly addressed in appendix G (both "conceptually", by highlighting the differences between both methods, and showing the advantages of the proposed SGDP and AdamP, but also empirically on ImageNet, where a fair comparison with proper hyper-parameter tuning has been performed).

The authors also reinforced their empirical investigation by reporting standard deviation of the results, which allows to better appreciate the performances of SGDP and AdamP. Finally, they also added the experiments with higher weight decay, showing that indeed 1e-4 was the best value.

With all these changes, I think the paper is good and I recommend its acceptance.

---

> ### Author Response · Authors · 2020-11-13
> **Author response (1/2)**
>
> Thank you for the high-quality expert review. We first address the concern regarding the comparison against an important prior work. We answer the rest of your questions, some of which may take days to get the experimental results. They will be updated as soon as the results arrive. In the meantime, we would be happy to discuss further, so please do not hesitate to comment and ask follow-up questions as necessary.
>
> **Comparison with "Cho & Lee, Riemannian approach to batch normalization" [1]**
> Thank you for pointing out an important reference, and sorry for missing a discussion about this work; we were not aware of this paper. Indeed, both [A] and ours are motivated from the scale invariance of certain parameters in a neural network, and both propose a solution that reduces the radial component of the optimization steps. However, our contributions are still not significantly eclipsed by this paper. Our optimization algorithm is different from that of [A], with a different set of theories to justify the methodologies. And this difference results in a practical edge for our optimizers.
>
> The crucial difference between [A] and ours is in the space where the optimization is performed. [A] performs the gradient steps on the Riemannian manifold. Ours project the updates on the tangent planes of the manifold. Thus, ours operates on the same Euclidean space where SGD and Adam operate. From a theory point of view, [A] has made contributions to the optimization theory on a Riemannian manifold. Our contributions are along a different axis: we focus on the norm growth when the updates are projected onto the tangent spaces (Sections 2.4, 3.1, and 3.2). We contribute present theoretical findings that are not covered by [A].
>
> From the practicality point of view, we note that changing the very nature of space from Euclidean to Riemannian requires users to find the sensible ranges for many optimization hyperparameters again. For example, [A] has “used different learning rates for the weights in Euclidean space and on Grassmann [Riemannian] manifolds” (page 7 of [A]), while in our case hyperparameters are largely compatible between scale-invariant and scale-variant parameters, for they are both accommodated in the same kind of space. We have shown that SGDP and AdamP outperform the SGD and Adam baselines with exactly the same optimization hyperparameters (Section E.2). The widely used Xavier or Gaussian initializations are no longer available in the spherical Riemannian manifold, necessitating changes in the code defining parameters and their initialization schemes (e.g. [A] has employed a dedicated initialization based on truncated normals. See https://github.com/MinhyungCho/riemannian-batch-normalization/blob/d1ac938ca5af8af1b7c1d4f708c1aacd2d8cbab9/gutils.py#L65). Finally, [A] requires users to manually register scale-invariant hyperparameters. This procedure is not scalable, as the networks nowadays are becoming deeper and more complex and the architectures are becoming more machine-designed than handcrafted. Our optimizers automatically detect scale invariances through the orthogonality test (eq.11), and users do not need to register anything by themselves. The sum of all the pain points above is an optimization algorithm that is not readily applicable in many practical scenarios, as hinted in their Github manual (https://github.com/MinhyungCho/riemannian-batch-normalization#to-apply-this-algorithm-to-your-model) where users are instructed to go through five convoluted steps to apply the algorithm. In stark contrast, users wanting to apply SGDP or AdamP only need to (1) pip install, (2) import, and (3) replace the “torch.optim.SGD(Adam)” with an “SGDP (AdamP)” class instantiation. No change in hyperparameters, design choices, or model codes is required.
>
> We recognize [A] as a great contribution to the field, and we are glad to confirm that another group of researchers have been motivated from a similar problem and have come up with a solution that shares certain similarities with ours. We believe that our contribution is not diminished by [A] though. Our solution is based on a different optimization space (Euclidean as opposed to Riemannian) with a different algorithm (projection onto the tangent space as opposed to moving along the manifold) and this provides significant practicality benefits for users.
>
> As an addition to the above points, we are conducting experiments on the toy examples (Section 3.2) and ImageNet to empirically compare ours against [A]. We will share the results as soon as they are out (ETA: November 18). We will revise the paper with the discussion above as well as the experiments that are being run.
>
> [A] Cho, Minhyung, and Jaehyung Lee. "Riemannian approach to batch normalization." Advances in Neural Information Processing Systems. 2017.

---

> > ### Author Response · Authors · 2020-11-13
> > **Author response (2/2)**
> >
> > **Description of Figure 1**
> > Thank you for finding out this glitch. The figure has been updated after the text. We will fix the text to match the figure.
> >
> > **Table 1 and Figures F with higher weight decay values (e.g. 3e-4 or 1e-3)**
> > Thanks for the suggestion. We are training ResNet18 with higher weight decay values (3e-4 and 1e-3) for Table 1 and Figure F. We will share the results as soon as they are available.
> >
> >
> > **Mean and standard deviation**
> >
> > The standard deviation for Tables 3, 5 and D.2 are as follows.
> >
> > - Table  3
> >
> >   |              |\| Music Tagging \||\| Music Tagging \||\| Keyword Spotting \||\| Sound Event Tagging \||
> >   |:------------:|:-------------:|:-------------:|:----------------:|:-------------------:|
> >   |              |    ROC-AUC    |     PR-AUC    |     Accuracy     |       F1 score      |
> >   |  Adam + SGD  |    ± 0.055    |    ± 0.025    |        -         |       -             |
> >   |     AdamW    |    ± 0.081    |    ± 0.189    |      ± 0.055     |       ± 0.394       |
> >   | AdamP (ours) |    ± 0.074    |    ± 0.269    |      ± 0.061     |       ± 0.833       |
> >
> > - Table 5
> >
> >   |              |\|   CUB   \||\|   CUB  \||\| Cars-196 \||\| Cars-196 \||\|  InShop \||\| InShop \||\|   SOP   \||\|   SOP  \||
> >   |:------------:|:-------:|:------:|:--------:|:--------:|:-------:|:------:|:-------:|:------:|
> >   |              | Triplet |   PA   |  Triplet |    PA    | Triplet |   PA   | Triplet |   PA   |
> >   |     AdamW    |  ± 0.91 | ± 0.24 |  ± 0.82  |  ± 0.09  |  ± 0.52 | ± 0.08 |  ± 0.71 | ± 0.04 |
> >   | AdamP (ours) |  ± 0.62 | ± 0.77 |  ± 1.35  |  ± 0.19  |  ± 0.82 | ± 0.02 |  ± 0.74 | ± 0.09 |
> >
> > - Table D.2
> >   Biased MNIST Unbiased acc. at ρ
> >
> >   |\|           ρ          \||\|  .999  \||\|  .997  \||\|  .995  \||\|  .990  \||
> >   |:------------:|:------:|:------:|:------:|:------:|
> >   |     Adam     | ± 6.04 | ± 0.48 | ± 0.70 | ± 0.48 |
> >   | AdamP (ours) | ± 1.73 | ± 1.31 | ± 2.16 | ± 0.40 |
> >
> >   9-Class ImageNet
> >
> >   |              |\| Biased \||\| UnBiased \||\| ImageNet-A \||
> >   |:------------:|:------:|:--------:|:----------:|
> >   |     AdamW    | ± 0.27 |  ± 0.40  |   ± 1.06   |
> >   | AdamP (ours) | ± 0.21 |  ± 0.27  |   ± 0.82   |
> >
> > For the other experiments, we have reported the averaged results of 3 independent trials (randomizing data shuffling for stochastic optimizers, as well as other random factors in data augmentation). However, the data has been lost due to the limited capacity of our shared servers. We are running new training sessions to recover the standard deviation. We will share the results and update the paper (ETA: November 18).

---

> > > ### Comment · AnonReviewer4 · 2020-11-16
> > > **Thank you for your response!**
> > >
> > > Dear authors,
> > >
> > > This is just a small comment to thank you for detailing the differences with [A], clarifying the text around Figure 1, and being in the process of updating the paper with all the comments I had!
> > >
> > > I am looking forward to review the updated document (at around November 20), as it looks like you are on track to address all the concerns I had about the paper!

---

> > > ### Author Response · Authors · 2020-11-21
> > > **Additional revision summary**
> > >
> > > We are pleased to announce that all experiments are completed and the paper has been revised accordingly.
> > >
> > > 1. **Standard deviation** We have performed experiments for standard deviation of all experiments in paper. The results show that the improvement of our optimizers is significantly larger than standard deviation in most cases. Please see **Section H.2** for details.
> > > 2. **Comparison with “Riemannian approach to batch normalization”** We compared our optimizer with “Riemannian approach to batch normalization”. The comparison includes theoretical and practical differences and experiments in the 3D toy example and ImageNet. Please see **Section G** of the revised paper for details.
> > > 3. **Baseline optimizers with high weight decay (1e-3, 3e-4)** As you commented, we conduct ImageNet experiments with higher weight decay (1e-3, 3e-4). As you expected, high weight decay does not improve the performance of the baseline optimizers. But, the results improve the completeness of our experiments. Please see **Section F.1** for details

---

### Official Review · AnonReviewer1 · 2020-10-27
**Review for AdamP: Slowing Down the Slowdown for Momentum Optimizers on Scale-invariant Weights**

**Rating:** 5
**Confidence:** 4

**Review:**

###################################################################

Summary:

This paper shows that momentum-based gradient descent optimizers reduce the effective step size in training scale-invariant models including deep neural networks normalized by batch normalization, layer normaliztion, instance normalization and group normalization. The authors then propose a solution that projects the update at each step in gradient descent onto the tangent space of the model parameters. Theoretical results are provided to show that this projection operator only adjusts the effective learning rate but does not change the effective update directions. Empirical results on various tasks are provided to justify the advantage of the proposed method over the baseline momentum-based (stochastic) gradient descent and Adam.

###################################################################

Reason for the Score:

Overall, this paper could be an interesting algorithmic contribution. However, there are relevant points needed to be clarified on the theory and experiments. My first main concern is that theoretically it is hard to justify that the proposed projection-based update yields smaller parameter norms than the baseline momentum-based update. My second main concern is some baseline results in the experiments do not match those in existing literature, and no error bars are provided in the empirical results even though the improvements of the proposed methods over the baseline methods are small.

Currently, I am leaning toward rejecting the paper. However, given additional clarifications on the two main concerns above in an author response, I would be willing to increase the score.

###################################################################

Strong points:

1. The paper points out a relevant issue in using normalization techniques such as batch normalization together with momentum-based optimization algorithms in training deep neural networks.

2. The paper provides experimental results on various tasks and datasets to demonstrate the advantage of the proposed method.

3. The paper is well-written with illustrative figures.

###################################################################

Weak points:

1. It is not clear to me that the proposed update in equation (12) yields smaller norms ||w_{t+1}|| than the momentum-based update in equation (8).  The parameters of the model evolve differently under these two update rules. Throughout the training, the update p_t in equation (11) is different from the update p_t in equation (8). As a result, it is hard to compare ||q_t|| in equation (12) and ||p_t|| plus all the terms ||p_k|| in equation (8).

2. The improvements of the proposed SGDP and AdamP over the baseline SGD and Adam are small across experiments, and thus error bars are needed to validate that these improvements are not due to randomness. However, no error bars are provided for the empirical results in the paper.

3. The reported baseline results for audio classification are worse than those reported in (Won et al., 2019).

4. The baseline results for adversarial robustness seems to be much higher than reported results in (Madry et al., 2018). Also why are the values of epsilon used in the paper quite small (80/255 and 4/255 vs. 8 in (Madry et al., 2018)).

###################################################################

Additional Concerns and Questions for the Authors:

1. Adam normalizes the gradient by its cumulative norm. This can help eliminate the small step size issue since the norms of the gradients become smaller during training. Can you provide a similar simulation as in Figure 3 but using Adam, AdamW, and AdamP?

2. What are the baseline results, reported in existing literature, on ImageNet for ResNet18 and ResNet50 trained with the cosine learning rate schedule in 100 epochs? Can you please link me to the previous papers that report those results? The paper you cite, (Loshchilov & Hutter, 2016), does not report those results.

3. In section 4.1, the authors say “For ResNet, we employ the training hyperparameters in (He, 2016)”. However, the training hyperparameters for ResNet used in the paper are not from (He, 2016). In (He, 2016), the models are trained for only 90 epochs without using cosine learning rate.

4. The proposed update is more expensive than the baseline momentum-update. The paper also reports that the proposed update incurs 8% extra training time on top of the baselines for ResNet18 on ImageNet classification while resulting in only small improvements over the baselines. It is needed to compare the proposed optimizer with the baseline momentum-based optimizer trained with more epochs and potentially with an additional learning rate decay.

###################################################################

Minor Comments that did not Impact the Score:

1. The paper proposes not only AdamP, but also SGDP. It is better if the authors remove AdamP in the title.

2. In figure 4, is the y-axis the test or train accuracy?

###################################################################

References:

Minz Won, Sanghyuk Chun, and Xavier Serra. Toward interpretable music tagging with self- attention. arXiv preprint arXiv:1906.04972, 2019.

Aleksander Madry, Aleksandar Makelov, Ludwig Schmidt, Dimitris Tsipras, and Adrian Vladu. Towards deep learning models resistant to adversarial attacks. In International Conference on Learning Representations (ICLR), 2018. URL https://openreview.net/forum?id= rJzIBfZAb.

Ilya Loshchilov and Frank Hutter. SGDR: Stochastic gradient descent with warm restarts. arXiv preprint arXiv:1608.03983, 2016.

Kaiming He, Xiangyu Zhang, Shaoqing Ren, and Jian Sun. Deep residual learning for image recognition. In Proceedings of the IEEE Conference on Computer Vision and Pattern Recognition (CVPR), 2016.


###################################################################

Post Discussion Score:

After reading the rebuttal from the author and the comments from other reviewers, I am still not clear if the proposed update in equation (12) yields smaller norms ||w_{t+1}|| than the momentum-based update in equation (8). However, the authors have addressed all of my other concerns. I decided to increase my score for this paper from 4 to 5.

---

> ### Author Response · Authors · 2020-11-13
> **Author response (1/3)**
>
> Thank you for the high-quality review and the extensive verification of experimental setups. We have answered most of your questions, except for the ones that require several days to finish training. They will be updated as soon as the results arrive. In the meantime, we would be happy to discuss further, so please do not hesitate to comment and ask follow-up questions as necessary.
>
> **It is not clear to me that the proposed update in equation (12) yields smaller norms ||w_{t+1}|| than the momentum-based update in equation (8).**
>
> We note that the size comparison of $\boldsymbol{p}_t$ and $\boldsymbol{q}_t$ is not presented in the current version; we will include it in the revision.
>
> We claim that $|| \boldsymbol{q}_t ||_2$ is always smaller than $|| \boldsymbol{p}_t ||_2$. This observation follows quickly from that fact that $\boldsymbol{q}_t$ is an output of a projection operation on $\boldsymbol{p}_t$. Since a projection mapping has eigenvalues either 0 or 1, it *always* decreases the norm of the input vector after transformation.
>
> Another way to see this is as follows:
> \\[  || \boldsymbol{q}_t ||_2 = || \boldsymbol{p}_t - (\hat{\boldsymbol{w}} \cdot \boldsymbol{p}_t) \hat{\boldsymbol{w}} ||_2 = \sqrt{\boldsymbol{p}_t \cdot \boldsymbol{p}_t - (\hat{\boldsymbol{w}} \cdot \boldsymbol{p}_t)^2} \leq || \boldsymbol{p}_t ||_2 \\]
>
>
> This inequality leads to the conclusion that:
> \\[  || {\boldsymbol{w}_t }||_2^2 + \eta^2 || {\boldsymbol{q}}_t ||_2^2 \leq || \boldsymbol{w}_t ||_2^2 + \eta^2 || {\boldsymbol{p}}_t ||_2^2 \leq || \boldsymbol{w}_t ||_2^2 + \eta^2 || {\boldsymbol{p}}_t ||_2^2 + 2 \eta^2 \sum_k \beta^{t-k}  ||{\boldsymbol{p}}_k||^2_2 \\]
>
> Thus, equation (12) yields smaller norms than equation (8). We will add these formulas in the paper.
>
> **The improvements of the proposed SGDP and AdamP over the baseline SGD and Adam are small across experiments, and thus error bars are needed to validate**
>
> The error bars for Tables 3, 5 and D.2 are as follows.
>
>
> - Table  3
>
>   |              |\| Music Tagging \||\| Music Tagging \||\| Keyword Spotting \||\| Sound Event Tagging \||
>   |:------------:|:-------------:|:-------------:|:----------------:|:-------------------:|
>   |              |    ROC-AUC    |     PR-AUC    |     Accuracy     |       F1 score      |
>   |  Adam + SGD  |    ± 0.055    |    ± 0.025    |        -         |       -             |
>   |     AdamW    |    ± 0.081    |    ± 0.189    |      ± 0.055     |       ± 0.394       |
>   | AdamP (ours) |    ± 0.074    |    ± 0.269    |      ± 0.061     |       ± 0.833       |
>
> - Table 5
>
>   |              |\|   CUB   \||\|   CUB  \||\| Cars-196 \||\| Cars-196 \||\|  InShop \||\| InShop \||\|   SOP   \||\|   SOP  \||
>   |:------------:|:-------:|:------:|:--------:|:--------:|:-------:|:------:|:-------:|:------:|
>   |              | Triplet |   PA   |  Triplet |    PA    | Triplet |   PA   | Triplet |   PA   |
>   |     AdamW    |  ± 0.91 | ± 0.24 |  ± 0.82  |  ± 0.09  |  ± 0.52 | ± 0.08 |  ± 0.71 | ± 0.04 |
>   | AdamP (ours) |  ± 0.62 | ± 0.77 |  ± 1.35  |  ± 0.19  |  ± 0.82 | ± 0.02 |  ± 0.74 | ± 0.09 |
>
> - Table D.2
>   Biased MNIST Unbiased acc. at ρ
>
>   |\|           ρ          \||\|  .999  \||\|  .997  \||\|  .995  \||\|  .990  \||
>   |:------------:|:------:|:------:|:------:|:------:|
>   |     Adam     | ± 6.04 | ± 0.48 | ± 0.70 | ± 0.48 |
>   | AdamP (ours) | ± 1.73 | ± 1.31 | ± 2.16 | ± 0.40 |
>
>   9-Class ImageNet
>
>   |              |\| Biased \||\| UnBiased \||\| ImageNet-A \||
>   |:------------:|:------:|:--------:|:----------:|
>   |     AdamW    | ± 0.27 |  ± 0.40  |   ± 1.06   |
>   | AdamP (ours) | ± 0.21 |  ± 0.27  |   ± 0.82   |
>
> For the other experiments, we have reported the averaged results of 3 independent trials (randomizing data shuffling for stochastic optimizers, as well as other random factors in data augmentation). However, the data has been lost due to the limited capacity of our shared servers. We are running new training sessions to recover the error bars. We will share the results and update the paper (ETA: November 18).

---

> > ### Author Response · Authors · 2020-11-13
> > **Author response (2/3)**
> >
> > **The reported baseline results for audio classification are worse than those reported in (Won et al., 2019).**
> >
> > Sorry for the confusion. The correct reference is (Won et al., 2020), while (Won et al., 2019) is a reference for the complex “mixture of Adam and SGD” (last paragraph of Section 4.3). We will accordingly fix the caption of Table 3 as “Audio classification. Results on three audio classification tasks with Harmonic CNN. **(Won et al., 2020)** ” to avoid the confusion.
> >
> > - Minz Won, Sanghyuk Chun, Oriol Nieto, and Xavier Serra. Data-driven harmonic filters for audio representation learning. In International Conference on Acoustics, Speech, and Signal Processing, 2020.
> > - Minz Won, Sanghyuk Chun, and Xavier Serra. Toward interpretable music tagging with self- attention. arXiv preprint arXiv:1906.04972, 2019.
> >
> > We reproduce the numbers based on the official implementation with the hyperparameters provided by the authors (Won et al., 2020). In MTAT ROC-AUC, (Won et al., 2020) have reported 91.41 and a cleaned-up version in a more recent paper [A] reports 91.27. Our numbers reproduce the second result 91.27.
> >
> > [A] Minz Won, Andres Ferraro, Dmitry Bogdanov, and Xavier Serra. Evaluation of CNN-based Automatic Music Tagging Models, SMC 2020
> >
> >
> > **The baseline results for adversarial robustness seems to be much higher than reported results in (Madry et al., 2018). Also why are the values of epsilon used in the paper quite small (80/255 and 4/255 vs. 8 in (Madry et al., 2018)).**
> >
> > We did not exactly follow the setting in (Madry et al., 2018). Below is the exact difference between our setting and Table 2 in (Madry et al., 2018):
> > * Our setting. Norm=Linf, eps=4/255, #iterations=10.
> > * (Madry et al., 2018) setting. Norm=Linf, eps=8/255, #iterations=20.
> >
> > As for the epsilon value, please note that there are two different ways of denoting the magnitude: (1) in real-number (floating point) and (2) in integer representation of images. (Madry et al., 2018) have used (2), while we choose to use (1). Thus, “8” in (Madry et al., 2018) corresponds to “8/255” in our paper.
> >
> > The technical differences notwithstanding, we fail to sympathize with the exact need to match our threat model with the one in (Madry et al., 2018). What is important in our evaluation is a fair evaluation: the same threat model is used for all baseline and our optimization methods. Under this setting, we have shown that our optimizer trains a more resilient model than does the baseline.
> >
> > Nonetheless, we have quickly obtained the adversarial robustness results with the same test-time threat model in (Madry et al., 2018) in the table below (see “re-evaluated” rows). Those models are still adversarially trained with the weak threat model, so we are running two more adversarial training sessions with the threat model with epsilon=8/255 in (Madry et al., 2018) (ETA: 19th Nov). The final results will be updated in the revision.
> >
> > |\|           Method           \||\|   train eps   \||\|          attack method          \||\|   attacked acc   \||\|               Note               \||
> > |:-------------------------:|:------------:|:---------------------------------:|:-----------------:|:------------------------------:|
> > | (Madry et al., 2018) | 8/255      |  PGD eps=8/255, iter=20  |  45.8%          | from Madry et al, 2018 |
> > | Adam                      | 4/255      |  PGD eps=8/255, iter=20  |  29.0%          | re-evaluated                 |
> > | AdamP                    | 4/255      |  PGD eps=8/255, iter=20  |  35.7%          | re-evaluated                 |
> > | Adam                       | 4/255      |  PGD eps=4/255, iter=10  |  56.6%          | reported in our paper   |
> > | AdamP                    | 4/255      |  PGD eps=4/255, iter=10  |  66.3%          | reported in our paper   |
> > | Adam                      | 8/255      |  PGD eps=8/255, iter=20  |         -            |    ETA: 19th Nov          |
> > | AdamP                    | 8/255      |  PGD eps=8/255, iter=20  |         -            |    ETA: 19th Nov          |
> >
> >
> > So far, with a stronger test-time threat model, we still have the same conclusion as before: AdamP outperforms Adam.

---

> > > ### Author Response · Authors · 2020-11-21
> > > **Additional revision summary**
> > >
> > > We are pleased to announce that all experiments are completed and the paper has been revised accordingly.
> > >
> > > 1. **Standard deviation** We have performed experiments for standard deviation of all experiments in the paper. The results show that the improvement of our optimizers is significantly larger than standard deviation in most cases. Please see **Section H.2** for details.
> > > 2. **Adversarial training for 8 in (Madry et al., 2018)** We perform adversarial training with epsilon=8/255 which is the same as in (Madry et al., 2018). In this setting, our AdamP shows significantly better performance than Adam optimizer. Please see **Section E.4.1** for details.
> > > 3. **3D toy for Adam optimizer** We conduct 3D toy examples for Adam series optimizer as your comment. The result is not very different from SGD, norm growth also occurs in Adam series, and our projection effectively prevents it. Please see **Section H.1** for details.
> > > 4. **Comparison in the same training budget** We also verified the performance of our optimizer on the same training budget. We trained our optimizer with 8% fewer epochs, which reduces performance, but confirms that our optimizer still outperforms the baseline optimizer. Please see **Section H.4** for details

---

> > ### Author Response · Authors · 2020-11-13
> > **Author response (3/3)**
> >
> > **Adam normalizes the gradient by its cumulative norm. This can help eliminate the small step size issue since the norms of the gradients become smaller during training. Toy example of Figure 3 for Adam series optimizers**
> >
> > The reduction in effective step sizes covered in this paper occurs due to the increase in weight norm. In other words, the denominator in eq (5) $  ||\Delta\widehat{\boldsymbol{w}}_t||_2= \frac{||\Delta\boldsymbol{w}_t||_2}{||\boldsymbol{w}_t||_2} $ increase and it reduces the effective step sizes. Although Adam may prevent the numerator from shrinking, the effect on the denominator is the same, so Adam does not solve the problem. We have performed the requested experiment on the example in Figure 3. Please find the results at the end of the Appendix after the paper revision.
> >
> >
> > **Where does the training setting of cosine learning rate schedule + 100 epochs come from? Cannot find the same settings and results in (Loshchilov & Hutter, 2016) or (He, 2016).**
> >
> > Our experimental setting (cosine learning rate and 100 epochs) for ImageNet is identical to that for RandAugment [B], with a smaller number of epochs and batch size, in order to reduce training time and perform more interesting experiments.
> > In summary, as the reviewer has rightly pointed out, we used the same learning rate, momentum, and weight decay as the ResNet training in [D], but a different number of epochs and learning rate schedules. We will describe the differences in greater detail in the revision. Sorry for the confusion.
> >
> > [B] Cubuk, Ekin D., et al. Randaugment: Practical automated data augmentation with a reduced search space. Proceedings of the IEEE/CVF Conference on Computer Vision and Pattern Recognition Workshops. 2020.
> > [C] Loshchilov, Ilya, and Frank Hutter. Decoupled Weight Decay Regularization. International Conference on Learning Representations. 2018.
> > [D] He, Kaiming, et al. Deep residual learning for image recognition. Proceedings of the IEEE conference on computer vision and pattern recognition. 2016.
> >
> >
> >
> > **Comparison with the baseline momentum-based optimizer trained with more (8%) epochs**
> > We are running the requested experiment. We will report the results as soon as they become available.
> >
> > **It is better if the authors remove AdamP in the title.**
> >
> > We agree that SGDP is also an important part of this paper. We have been slightly favouring AdamP over SGDP because many researchers and engineers who have used our optimizers have found that AdamP has often shown better results (see Table 1). This has been reflected in the title. However, it seems a better choice to not narrow down the focus of title to AdamP to do justice for SGDP. In the final version, we will consider deleting AdamP from the title.
> >
> > **In figure 4, is the y-axis the test or train accuracy?**
> > It represents the test set accuracy. We will update the y-axis label.

---

> ### Author Response · Authors · 2020-11-16
> **Corrections to the author response**
>
> **Corrections to the response to the size comparison between $\boldsymbol{p}_t$ and $\boldsymbol{q}_t$.**
>
> After carefully reading your review again, we realize that we have misunderstood your question here. We thought you were wondering about the size comparisons between the pre-projection update $\boldsymbol{p}_t$ and the projected update $\boldsymbol{q}_t$ at a single time snapshot $t$. We have answered that the projection indeed decreases the weight norm increase at a particular time step $t$ because a projection always contracts the norm. However, your question was asking whether the weight norm increase in the sequence of steps $0,\ldots,t$ is reduced by a sequential application of projection, asking for the exact size comparison between equation (8) and (12). We answer your real question below. Sorry for a delayed realization.
>
> It is not trivial to write down a clean formula for the norm size increases $\| \boldsymbol{w}_t \|_2^2 - \| \boldsymbol{w}_0\|_2^2$ for GD with our projection (equation (12)) because of the intricate interdependency among the updates $\boldsymbol{q}_t, \cdots, \boldsymbol{q}_1$, gradients $\nabla f(\boldsymbol{w}_t), \cdots, \nabla f(\boldsymbol{w}_0)$, and the weights $\Pi_\boldsymbol{w_t}, \cdots, \Pi_\boldsymbol{w_0}$. Moreover, as the reviewer is already aware, the norm size increase will depend on the function gradients along the optimization paths that will differ for the two optimizers. As a result, the relationship between the norm increases between the two optimizers will further depend on the geometry of the function, among other things. A direct comparison will only be possible with strong assumptions on the smoothness of $f$, which will then make the theoretical results less relevant in real life.
>
> Instead of trying to build a theoretical argument based on a set of unrealistically strong assumptions, we point to the fact that our optimizers do reduce the norm growth in practice. See Section 3.2, where we show both in toy and real-world examples (ImageNet) that models trained with projections have smaller weight norm increases throughout the training. In the revision, we will explain that the direct, analytical comparison is difficult and that we instead provide empirical evidence to support the claim that our method reduces the weight norm increase.

---

> ### Comment · AnonReviewer1 · 2020-11-22
> **Reply to the Author's Response from AnonReviewer1**
>
> Thank you for your responses. I have gone through the changes and also read the other reviews. Below is my main concern. Except for this, I am happy with the author’s responses to my questions in the rebuttal.
>
>
> On Question: It is not clear to me that the proposed update in equation (12) yields smaller norms ||w_{t+1}|| than the momentum-based update in equation (8).
>
>
> As I mentioned in my review and the authors also re-confirmed in their response, p_t are not identical between Eq. (8) and Eq. (12). Thus, it cannot be theoretically claimed that q_t in Eq. (12) is smaller than p_t in Eq. (8). The authors have provided empirical results to verify this claim in the paper and the revision (Figure 3, F.1, F.2, F.3, F.4, H.1).  Figure 3 shows that GD + the proposed projection yields a smaller norm of parameters throughout the training on the toy 3D scale-invariant Rosenbrock problem. However, the results from Figure F.1, F.2, F.3, F.4, which are for the ImageNet classification task, raise concern. In particular, it seems that the baseline SGD with wd=1e-4 yields smaller norms of parameters than the SGD + the proposed projection (SGDP) in the later part of the training. This reflects in the top-1 accuracies of SGD (wd = 1e-4) and the proposed SGDP. Both methods yield quite similar top-1 accuracies for the ImageNet classification task. Furthermore, in Figure H.1 which studies 3D scale-invariant Rosenbrock with Adam optimizers, Adam + weight decay yields a smaller norm of parameters than Adam + the proposed projection.
>
>
> Comments that did not Impact the Score:
>
> Given the provided error bars for Table 3, AdamP does not necessarily yield better PR-AUC for the music tagging task compared to the baseline AdamW.
>
> Given the provided error bars for Table 5, AdamP does not necessarily yield better results CUB and Cars-196 (Triplet), compared to the baseline AdamW.
>
> However, I am happy with the standard deviation that the authors provided since it empirically confirms the advantage of SGDP and AdamP over the baselines, even though most of the advantages are small except for the real-world bias robustness results (table D.2).

---

### Official Review · AnonReviewer2 · 2020-10-27
**Good insights, promising results, but missing important related work**

**Rating:** 6
**Confidence:** 4

**Review:**

This paper points out that momentum in GD optimizers results in a far more rapid reduction in effective step sizes for scale-invariant weights.  To solve the problem, two algorithms called SGDP and AdamP are proposed, which project the updates to tangent space of the parameter. Experiments on several tasks including image classification, language modeling, etc show the effectiveness of the proposed algorithms. The idea to study the integration of BN and momentum is interesting. The analyses and proposed algorithms provide guidance to practitioners.

Questions:

(1) What is the difference between the proposed algorithm and the algorithm in paper "Cho & Lee, Riemannian approach to batch normalization"? Eq.(10) in your paper is exactly the same with Eq.(6) in [Cho & Lee]. It is an important related work which should be cited and compared with.

(2) What does "lies on the geodesic" mean in Proposition 3.1? What is the relation between Proposition 3.1 and the convergence of the algorithm?

(3) It is not clear which results are proposed in related work and which results are proposed in this paper. Is Lemma 2.1 a new result? If not, it needs a citation.

(4) The theory part analyzes the effect of momentum, while the experiments shows the effect of weight decay. How does weight decay influence the norm growth theoretically? Why not conduct experiments with varying momentum coefficient?

(5) The theory part shows the negative effect of momentum. Does it mean that "SGD is a better choice than momentum SGD"? Momentum SGD is a standard algorithm to train deep neural networks with BN in practice but not vanilla SGD. How to explain it?

(6) What is the additional computational cost of the proposed algorithms compared with the baselines?

---

> ### Author Response · Authors · 2020-11-13
> **Author response (1/2)**
>
> Thank you for the high-quality review with spot-on questions. We include the requested comparison against the important missing prior work. We answer the rest of your questions as thoroughly as possible. We would be happy to discuss further, so please do not hesitate to comment and ask follow-up questions.
>
> **Difference from "Cho & Lee, Riemannian approach to batch normalization" [A] ?**
> Thank you for pointing out an important reference, and sorry for missing a discussion about this work; we were not aware of this paper. Indeed, both [A] and ours are motivated from the scale invariance of certain parameters in a neural network, and both propose a solution that reduces the radial component of the optimization steps. However, our contributions are still not significantly eclipsed by this paper. Our optimization algorithm is different from that of [A], with a different set of theories to justify the methodologies. And this difference results in a practical edge for our optimizers.
>
> The crucial difference between [A] and ours is in the space where the optimization is performed. [A] performs the gradient steps on the Riemannian manifold. Ours project the updates on the tangent planes of the manifold. Thus, ours operates on the same Euclidean space where SGD and Adam operate. From a theory point of view, [A] has made contributions to the optimization theory on a Riemannian manifold. Our contributions are along a different axis: we focus on the norm growth when the updates are projected onto the tangent spaces (Sections 2.4, 3.1, and 3.2). We contribute present theoretical findings that are not covered by [A].
>
> From the practicality point of view, we note that changing the very nature of space from Euclidean to Riemannian requires users to find the sensible ranges for many optimization hyperparameters again. For example, [A] has “used different learning rates for the weights in Euclidean space and on Grassmann [Riemannian] manifolds” (page 7 of [A]), while in our case hyperparameters are largely compatible between scale-invariant and scale-variant parameters, for they are both accommodated in the same kind of space. We have shown that SGDP and AdamP outperform the SGD and Adam baselines with exactly the same optimization hyperparameters (Section E.2). The widely used Xavier or Gaussian initializations are no longer available in the spherical Riemannian manifold, necessitating changes in the code defining parameters and their initialization schemes (e.g. [A] has employed a dedicated initialization based on truncated normals. See https://github.com/MinhyungCho/riemannian-batch-normalization/blob/d1ac938ca5af8af1b7c1d4f708c1aacd2d8cbab9/gutils.py#L65). Finally, [A] requires users to manually register scale-invariant hyperparameters. This procedure is not scalable, as the networks nowadays are becoming deeper and more complex and the architectures are becoming more machine-designed than handcrafted. Our optimizers automatically detect scale invariances through the orthogonality test (eq.11), and users do not need to register anything by themselves. The sum of all the pain points above is an optimization algorithm that is not readily applicable in many practical scenarios, as hinted in their Github manual (https://github.com/MinhyungCho/riemannian-batch-normalization#to-apply-this-algorithm-to-your-model) where users are instructed to go through five convoluted steps to apply the algorithm. In stark contrast, users wanting to apply SGDP or AdamP only need to (1) pip install, (2) import, and (3) replace the “torch.optim.SGD(Adam)” with an “SGDP (AdamP)” class instantiation. No change in hyperparameters, design choices, or model codes is required.
>
> We recognize [A] as a great contribution to the field, and we are glad to confirm that another group of researchers have been motivated from a similar problem and have come up with a solution that shares certain similarities with ours. We believe that our contribution is not diminished by [A] though. Our solution is based on a different optimization space (Euclidean as opposed to Riemannian) with a different algorithm (projection onto the tangent space as opposed to moving along the manifold) and this provides significant practicality benefits for users.
>
> As an addition to the above points, we are conducting experiments on the toy examples (Section 3.2) and ImageNet to empirically compare ours against [A]. We will share the results as soon as they are out (ETA: November 18). We will revise the paper with the discussion above as well as the experiments that are being run.
>
> [A] Cho, Minhyung, and Jaehyung Lee. "Riemannian approach to batch normalization." Advances in Neural Information Processing Systems. 2017.

---

> > ### Author Response · Authors · 2020-11-13
> > **Author response (2/2)**
> >
> > **What does "lies on the geodesic" mean in Proposition 3.1? What is the relation between Proposition 3.1 and the convergence of the algorithm?**
> > A geodesic (great circle) on a sphere is the equivalent of a line on a Euclidean space. Like a line in a Euclidean space, a geodesic is uniquely defined by two points (that are not antipodal), and it represents the “direction” on a sphere. Thus, the phrase “a point x lies on the geodesic defined by y and z on a sphere” signifies that the direction defined by any pair of the three, (x,y), (y,z) and (z,x), are identical (they are on the same line). In the context of Proposition 3.1, this means that the direction of the optimization update, from the spherical-space point of view, is identical before and after our projection algorithm. This is a desirable property, as it assures that our projection algorithm is not arbitrarily altering the update vector to an irrelevant direction. Based on this proposition, we argue that the convergence properties of the baseline optimizer (without projection) are inherited by our optimizer (with projection).
> >
> > **It is not clear which results are proposed in related work and which results are proposed in this paper. Is Lemma 2.1 a new result? If not, it needs a citation.**
> > Materials in Section 2.3 (including Lemma 2.1) are the background results based on [2] and [3]. Materials in Section 2.4 and Section 3 are our original work. We will make this clearer in the paper revision.
> >
> > **The theory part analyzes the effect of momentum, while the experiments shows the effect of weight decay. How does weight decay influence the norm growth theoretically? Why not conduct experiments with varying momentum coefficient?**
> > Since our optimizer is designed to regulate the growth of weight norms, finding a good weight decay hyperparameter (L2 regularization on weights) is arguably a simple baseline that may achieve the same effect. Thus, we have treated the comparison of our optimizer against various weight decay options with gravity. We have verified that our scheme handles the growth of weight norms more gracefully (Figure 3 & Table 1) and achieves better model accuracies (Table 1). We agree that the momentum coefficient is also deeply related to our proposed method. However, we have considered the exploration of the coefficient relatively less significant, since the value 0.9 is used in almost all practical tasks. Nonetheless, we agree that exploration beyond the standard value 0.9 will be an interesting addition to our paper. We will report experimental results on this.
> >
> > **The theory part shows the negative effect of momentum. Does it mean that "SGD is a better choice than momentum SGD"?**
> > We do acknowledge the benefits of momentum, as argued at the beginning of Section 2.4: “Momentum is designed to accelerate the convergence of gradient-based optimization by letting [the weights] escape high-curvature regions and cope with small and noisy gradients”. We do point out the unwanted side-effect of momentum, namely the excessive growth of weight norms, but that is not to say that momentum SGD is worse than the momentum-less SGD. Instead, we have proposed the momentum+projection solution to reduce the unwanted side-effect while “retaining the benefits of momentum” (Section 3.1). We will make this argument clearer in the paper.
> >
> > **What is the additional computational cost of the proposed algorithms compared with the baselines?**
> > Our method increases the training time by 8% for ResNet18 on ImageNet. This is noted in the last line of section 3.1.

---

> > > ### Author Response · Authors · 2020-11-21
> > > **Additional revision summary**
> > >
> > > We are pleased to announce that all experiments are completed and the paper has been revised accordingly.
> > >
> > > 1. **Comparison with “Riemannian approach to batch normalization”** We compared our optimizer with “Riemannian approach to batch normalization”. The comparison includes theoretical and practical differences and experiments in the 3D toy example and ImageNet. Please see **Section G** of the revised paper for details.
> > > 2. **Analysis for momentum coefficient** As your recommendation, we analyze our optimizer for various momentum coefficients through experiments in ImageNet. The analysis is included in **Section H.3** of the revised paper.

---

### Official Review · AnonReviewer3 · 2020-10-27
**Rewiew**

**Rating:** 6
**Confidence:** 2

**Review:**

Summary: Based on the assumption that the rapidly decrease step size \delta \omega leads to a solve effective convergence of \omega. This paper proposes to use  a projection step to remove the radial component, and thus reduce the norm of training parameters, and faster the training procedure. The experiments look good to me while the derivation of this paper based on many assumptions and conjectures.   I tend to accept this paper at this time. However, I am not an expert at this area I will not be sad if this paper is reject by other reviews.

minor problems:
1) Can the author explain more on how to derive eq.(4) ?

---

> ### Author Response · Authors · 2020-11-13
> **Author response**
>
> Thank you for spending time reading a paper out of your domain of expertise. We benefit a lot from reviewers from other domains, as we may learn how to make the paper interesting for a broader audience. We answer your question below. Please do not hesitate to ask for further clarifications for any part of the paper, should there be points that are not so obvious.
>
> **How to derive eq.(4)**
> We start from the observation that if f(w) is scale invariant with respect to w, then
> \\[ \frac{\partial f(c\boldsymbol{w})}{\partial c}=0. \\]
> This follows from the fact that the rate of change along w is always zero. Now, by the multivariate chain rule, we compute
> \\[ \frac{\partial f(c\boldsymbol{w})}{\partial c}=
> \sum_i \left[ \frac{\partial f}{\partial v_i} \frac{\partial v_i}{\partial c} \right]_{v = c\boldsymbol{w}}  \\]
> where $\boldsymbol{v}=c\boldsymbol{w}$. We continue the derivation as follows:
> \\[ \sum_i \left[ \frac{\partial f}{\partial v_i} \frac{\partial v_i}{\partial c} \right]_{v = c\boldsymbol{w}} =
> \sum_i \nabla f(\boldsymbol{v})_i w_i =
> \sum_i \nabla f(\boldsymbol{w})_i w_i =
> \boldsymbol{w}^\top \nabla f(\boldsymbol{w}).\\]
> Hence, it follows that the scale-invariance leads to the orthogonality between gradient and the radial direction:
> \\[  \boldsymbol{w}^\top \nabla f(\boldsymbol{w})= 0. \\]

---

### Public Comment · ~Juntang_Zhuang1 · 2020-11-13
**Missing reference to a related work and a few questions**

Hi, thanks for the nice work, this is very inspiring. I want to point to our work [1], which proposes an optimizer based on the change in gradient, would really appreciate it if you could briefly discuss.

I have a few question, appreciate it if you could discuss. 1) If I understand correctly, AdamP and SGDP is basically changing the direction of the weight but keeps the norm constant, does this imply that it can be combined with any other gradient-based methods? 2) This regularization depends on the assumption that the weight should be close to 0, which is often true in deep learning with over-parameterization and normalization, is it suitable for conventional optimization where the weights is not zero-centered? 3) Is it suitable for RNN, which often suffers from exploding or vanishing gradient?

Thanks a lot in advance, just curious if you could clarify these. In fact I think this work is very inspiring and supported with many experiments.

[1] Zhuang, Juntang, et al. "AdaBelief Optimizer: Adapting Stepsizes by the Belief in Observed Gradients." Advances in Neural Information Processing Systems 33 (2020).

---

> ### Author Response · Authors · 2020-11-15
> **Answers for your questions**
>
> Thank you for your interest in our paper and for introducing an interesting related work. AdaBelief is expected to be presented at NeurIPS 2020 next month. After your paper is published, we will consider adding a short discussion on your paper in the camera-ready version.
>
> **Can it be combined with any other gradient-based methods?**
> We believe our optimizer can be applied to most optimizers. After all, our algorithm is but a simple projection step after the base optimizer’s update. We believe it makes a lot of sense to apply ours on AdaBelief as well.
>
> **This regularization depends on the assumption that the weight should be close to 0. Is it suitable for conventional optimization where the weights is not zero-centered?**
> We do not require the weights to be close to zero or zero-centered.
> The only assumption for our method is that there exist scale-invariant parameters in the model. But many modern deep networks are built upon many normalization layers, and thus scale-invariant parameters. This is why we argue that our optimizer has a wide applicability. This assumption is also applied to conventional optimization problems other than deep learning. If the problem has a scale-invariant variable like our toy case (Figure 1, Figure 3), our optimizer can be used for the problem.
>
> **Is it suitable for RNN?** If you use  a normalization layer in the RNN, or introduce any other forms of scale invariance for weights, then yes. Our optimizer does not have much to do with exploding or vanishing gradients.

---

> > ### Public Comment · ~Juntang_Zhuang1 · 2020-11-16
> > **Thanks for your answer**
> >
> > Thanks for answering my questions. Regarding the scale-invariant parameters, if part of the parameters are scale-invariant, other parts are not, in this case the projection can only be applied on the scale-invariant part? For example in Fig 3, if not projecting (x,y) onto a sphere, instead direct work on the 2D plane, the projection is not feasible. Or if the parameters are (x,y,z), projecting (x,y) onto a sphere, but not z, in this case can your method be applied on z?

---

> > > ### Author Response · Authors · 2020-11-17
> > > **Partial scale-invariant case**
> > >
> > > **if part of the parameters are scale-invariant, other parts are not, in this case the projection can only be applied on the scale-invariant part?**
> > >
> > > If we know which part of the parameter is scale-invariant, we can change our scale-invariant detection and projection operation to cope with that. In the case of deep networks, the scale-invariant is mostly caused by the normalization layer (BatchNorm, LayerNorm etc.). Therefore, we designed the optimizer to correspond to the scale-invariant in each layer or channel, which can be considered as parts of the parameters of the entire network. There are a lot of ways to divide parameters into parts, and it is impractical to respond to all of them. However, we believe that most scale-invariants can be handled by modifying our optimizer based on the analysis of scale-invariants. Answering your example, if we know that (x, y) is scale-invariant, we can tune our optimizer. However, if our optimizer is used without tuning, it will not be able to respond to this scale-invariant because it is not the channel-wise or layer-wise scale-invariant.

---

### Author Response · Authors · 2020-11-15
**Paper revision plan**

We'd like to share our paper revision and experiments plans.  All the changes are highlighted in pink.

- First revision (Updated)
  - Text revision according to the review (All reviewers) : **Highlighted in pink**
  - Standard deviation update (Table 3, 5, D.2) (R#1, R#4) : **Appendix (Section. H.2)**
  - Toy example (Figure 3) for Adam series optimizers (R#1) : **Appendix (Section. H.1)**
- Second revision (Updated)
  - Standard deviation update (Table 2, 4) (R#1, R#4): **Appendix (Section. H.2)**
  - Theoretical comparison with “Riemannian approach to batch normalization” (R#2, 4): **Appendix (Section. G.1)**
  - Toy example with “Riemannian approach to batch normalization” (R#2, 4): **Appendix (Section G.1 and Figure. G.1, G.2)**
- Third revision (Updated)
  - Adversarial training results with epsilon=8/255 in (Madry et al., 2018) (R#1) **Appendix (Section E.4.1 and Table E.2)**
  - ImageNet experiments with “Riemannian approach to batch normalization” (R#2, 4)  **Appendix (Section G.1 and Table G.1)**
  - Comparison results with same computation cost (8% less epoch) (R#1) **Appendix (Section H.4 and Table H.7)**
  - Experiments for momentum coefficient (R#2) **Appendix (Section H.3 and Table H.6)**
  - ImageNet for high wd (1e-3, 3e-4) (R#4) **Appendix (Section F.1 and Figure F.5, F.6)**

Third revision of our paper has been uploaded. We are pleased to announce that all our experiments have been completed and the paper has been revised accordingly.

---

### Decision · Program_Chairs · 2021-01-07
**Final Decision**

**Decision:**

Accept (Poster)

**Comment:**

Clarity: The paper is well-written with illustrative figures.

Originality: The originality of the paper is relatively restricted, mainly due to the resemblance with the work [1]. However, there are important differences, that the authors nicely pointed out, and we encourage them to include these in the final version of the paper.

Significance: The paper points out a relevant issue in using normalization techniques such as batch normalization together with momentum-based optimization algorithms in training deep neural networks. While the paper could be considered "another algorithms for training NNs", the papers illustrates nicely the main arguments, and is backed up with more than sufficient experimental results.

Main pros:
- In the main pros, AC and reviewers admit the phenomenal job in responding to reviewers' questions and requests
- The paper provides experimental results on various tasks and datasets to demonstrate the advantage of the proposed method.
- After the reviews, The authors also reinforced their empirical investigation by reporting standard deviation of the results, which allows to better appreciate the performances of SGDP and AdamP. Finally, they also added the experiments with higher weight decay, showing that indeed 1e-4 was the best value.

Main cons:
- One reviewer requires more explanation why the proposed update in equation (12) yields smaller norms ||w_{t+1}|| than the momentum-based update in equation (8).